Isthminia panamensis, a new fossil inioid (Mammalia, Cetacea) from the Chagres Formation of Panama and the evolution of ‘river dolphins’ in the Americas

Pyenson Nicholas D. 1 2 pyensonn@si.edu
Vélez-Juarbe Jorge 3 4
Gutstein Carolina S. 1 5
Little Holly 1
Vigil Dioselina 6
O’Dea Aaron 6
1 Department of Paleobiology, National Museum of Natural History, Smithsonian Institution , Washington, DC , USA
2 Departments of Mammalogy and Paleontology, Burke Museum of Natural History and Culture , Seattle, WA , USA
3 Department of Mammalogy, Natural History Museum of Los Angeles County , Los Angeles, CA , USA
4 Florida Museum of Natural History, University of Florida , Gainesville, FL , USA
5 Comisión de Patrimonio Natural, Consejo de Monumentos Nacionales , Santiago , Chile
6 Smithsonian Tropical Research Institute , Balboa , Republic of Panama
Young Mark
Electronic publication date: 2015 Sep 1
Publication date: 2015
Volume: 3
Electronic Location ID: e1227
Received 2015 Apr 27; Accepted 2015 Aug 13
Copyright: © 2015 Pyenson et al.
Copyright year: 2015
Copyright holder: Pyenson et al.
License: This is an open access article distributed under the terms of the Creative Commons Attribution License, which permits unrestricted use, distribution, reproduction and adaptation in any medium and for any purpose provided that it is properly attributed. For attribution, the original author(s), title, publication source (PeerJ) and either DOI or URL of the article must be cited.
License URL: https://creativecommons.org/licenses/by/4.0/

Keywords: River dolphins, Cetacea, Panama, Fossil record, Evolution, Neogene, Inioidea, Amazonia

Funding: Smithsonian Institution NSF EAR Postdoctoral Fellowship 1249920 NSF PIRE 0966884 CONICYT National System of Investigators (SNI) NSF Collaborative Research award 1325379 The research of NDP was funded by the Smithsonian Institution, its Remington Kellogg Fund, and with support from the Basis Foundation. This project was also partially funded by an NSF EAR Postdoctoral Fellowship (1249920) to JVJ and NSF PIRE (0966884). Funding from CONICYT, Becas Chile, Departamento de Postgrado y Postítulo of the Vicerrectoría de Asuntos Académicos of Universidad de Chile supported CSG. This research was also supported financially by the National System of Investigators (SNI) of the National Secretariat for Science, Technology and Innovation of Panama (SENACYT) and a NSF Collaborative Research award (1325379) to AO. The funders had no role in study design, data collection and analysis, decision to publish, or preparation of the manuscript.

==============================
In contrast to dominant mode of ecological transition in the evolution of marine mammals, different lineages of toothed whales (Odontoceti) have repeatedly invaded freshwater ecosystems during the Cenozoic era. The so-called ‘river dolphins’ are now recognized as independent lineages that converged on similar morphological specializations (e.g., longirostry). In South America, the two endemic ‘river dolphin’ lineages form a clade (Inioidea), with closely related fossil inioids from marine rock units in the South Pacific and North Atlantic oceans. Here we describe a new genus and species of fossil inioid, Isthminia panamensis, gen. et sp. nov. from the late Miocene of Panama. The type and only known specimen consists of a partial skull, mandibles, isolated teeth, a right scapula, and carpal elements recovered from the Piña Facies of the Chagres Formation, along the Caribbean coast of Panama. Sedimentological and associated fauna from the Piña Facies point to fully marine conditions with high planktonic productivity about 6.1–5.8 million years ago (Messinian), pre-dating the final closure of the Isthmus of Panama. Along with ecomorphological data, we propose that Isthminia was primarily a marine inhabitant, similar to modern oceanic delphinoids. Phylogenetic analysis of fossil and living inioids, including new codings for Ischyrorhynchus, an enigmatic taxon from the late Miocene of Argentina, places Isthminia as the sister taxon to Inia, in a broader clade that includes Ischyrorhynchus and Meherrinia, a North American fossil inioid. This phylogenetic hypothesis complicates the possible scenarios for the freshwater invasion of the Amazon River system by stem relatives of Inia, but it remains consistent with a broader marine ancestry for Inioidea. Based on the fossil record of this group, along with Isthminia, we propose that a marine ancestor of Inia invaded Amazonia during late Miocene eustatic sea-level highs.

Introduction

In the evolution of marine mammals, the dominant mode of their ecological transitions (sensu Vermeij & Dudley, 2000) is the iterative adaptation to marine life from terrestrial ancestry (Thewissen & Williams, 2002; Gingerich, 2005; Kelley & Pyenson, 2015). However, the direction of this ecological transition is not exclusively from land to sea: throughout the late Cenozoic, several lineages of cetaceans and pinnipeds have evolved exclusively freshwater lifestyles from a marine ancestry (Hamilton et al., 2001; Pyenson, Kelley & Parham, 2014). Among cetaceans, the group of extant ‘river dolphins’ are the best exemplars of this ecological mode. This non-monophyletic (i.e., paraphyletic or possibly polyphyletic) group traditionally includes four different living species: Platanista gangetica (Lebeck, 1801); Lipotes vexillifer Miller, 1918, Inia geoffrensis (Blainville, 1817), and Pontoporia blainvillei (Gervais & d’Orbigny, 1844). These species all show broad morphological similarities, including longirostral skulls and jaws, reduced orbits, flexible necks, and broad, paddle-shaped flippers (Geisler et al., 2011). Notably, this assemblage of broadly convergent taxa has a biogeographic distribution across different freshwater river systems of South Asia and South America, and in estuarine and coastal waters of the latter as well.

While work for most of 20th century implied or proposed that the ‘river dolphins’ were all most closely related to one another (e.g., Simpson, 1945), the advent of molecular phylogenies clarified that these lineages are not all directly related to one another (see Geisler et al., 2011 for a useful summary), although both molecular and morphological analyses consistently group the two South American genera, Inia and Pontoporia, as sister taxa (Inioidea sensu Muizon, 1988a). Lipotes, which was endemic to the Yangtze River of China and is likely extinct (Turvey et al., 2010), may be the sister taxon to Inioidea (see Geisler et al., 2011), although all molecular studies (e.g., Messenger & McGuire, 1998; Hamilton et al., 2001; Nikaido et al., 2001; Geisler & Sanders, 2003; Arnason, Gullberg & Janke, 2004; May-Collado & Agnarsson, 2006; McGowen, Spaulding & Gatesy, 2009; Steeman et al., 2009; Geisler et al., 2011) and combined molecular and morphological analyses (Geisler et al., 2011; Gatesy et al., 2013) place Lipotes within Delphinida (i.e., Inioidea + Delphinoidea sensu Muizon, 1988a), and furthermore place Platanista outside of Delphinida. Lipotes and Platanista have only been grouped together in analyses using purely morphological datasets (e.g., Geisler & Sanders, 2003).

With restricted distributions, serious conservation threats, and relatively low taxonomic richness compared with other odontocete clades, the evolutionary history of ‘river dolphins’ remains a topic of perennial interest (Cassens et al., 2000; Hamilton et al., 2001; Nikaido et al., 2001; Pyenson, 2009; Ruiz-Garcia & Shostell, 2010; Turvey et al., 2010; Geisler et al., 2011). The fossil record of South Asian ‘river dolphins’ is poor, with no taxa reported from undisputable remains (e.g., Prolipotes yujiangensis Zhou, Zhou & Zhao, 1984 is known only from an isolated mandible that cannot be clearly diagnosed). By contrast, fossil South American ‘river dolphins’ have been reported from Neogene rocks of the continent since the 1850s (Cozzuol, 1996). The majority of these fossil taxa have been assigned to the traditional taxonomic groups of either Iniidae or Pontoporiidae, based on diagnostic features of the face and vertex (Muizon, 1988a), and include taxa (e.g., Pontistes rectifrons Burmeister, 1885, Pliopontos littoralis Muizon, 1983, Brachydelphis mazeasi Muizon, 1988b) known from marine rocks units of middle Miocene through Early Pliocene age in Argentina, Peru, Chile, and elsewhere (Muizon, 1984; Muizon, 1988b; Cozzuol, 1996; Gutstein et al., 2009; Lambert & de Muizon, 2013; Gutstein et al., 2014a). Recently, Bianucci et al. (2013) reported an isolated periotic with diagnostic features of Platanistinae (today limited to South Asia) from the Peruvian Amazon Basin of Laventan South American Land Mammal Age. This finding is striking for its disjunct biogeographic occurrence, relative to living Platanista in South Asia, but it is consistent with the widespread distribution of fossil platanistoids reported elsewhere in the world from late Paleogene through Neogene rocks along the coasts of the South and North Pacific and the North Atlantic oceans (Fordyce, 2009).

Similarly, the fossil record of inioids extends well beyond South America (Fig. 1). Fossil pontoporiids have been described from shallow marine and estuarine strata of early late Miocene to Early Pliocene age from the Atlantic coast of North America, including Maryland, Virginia, North Carolina and Florida (Morgan, 1994; Whitmore, 1994; Godfrey & Barnes, 2008; Gibson & Geisler, 2009; Geisler, Godfrey & Lambert, 2012). Along the Atlantic coast of Europe, Protophocaena minima Abel, 1905 from shallow marine Miocene of the Netherlands, is now recognized as a pontoporiid (Lambert & Post, 2005) based on additional cranial and periotic material from the Miocene of Belgium and the Netherlands. Pyenson & Hoch (2007) reported pontoporiids (cf. Pontistes sp. and indeterminate Pontoporiidae) from the marine Gram Formation in Denmark, which is early late Miocene age. To date, no fossil pontoporiids have been described from the North Pacific Ocean. The two species of Parapontoporia Barnes, 1984, which are well known from abundant Mio-Pliocene localities in northern and southern California (Boessenecker & Poust, 2015), are not pontoporiids, but belong in a clade with Lipotes (Geisler, Godfrey & Lambert, 2012), although Parapontoporia is sometimes also grouped with Platanista, Lipotes and Ischyrorhynchus vanbenedeni Ameghino, 1891 (see Aguirre-Fernández & Fordyce, 2014). Historically, fossils referred to Iniidae include a variety of taxa (e.g., Goniodelphis hudsoni Allen, 1941; Ischyrorhynchus), supplementing the existing data showing a much broader geographic extent for inioids in the fossil record than today (Fig. 1). These fossil occurrences thus raise the question of how Inioidea evolved, and the evolutionary scenarios that led to their current distribution. Our description herein of a new genus and new species of Inioidea from the late Miocene of Panama, based on substantially more osteological material than most fossil inioids, provides new insight into the evolutionary scenarios under which this group evolved in South America, including the timing and mode of major ecological transitions.

Figure 1 Map of fossil and living Inioidea.

Global map of living and fossil inioids, projected onto an orthographic globe, centered on 15°N, 45°W. Extant distributions of Inia geoffrensis (teal and black waterways) and Pontoporia blainvillei (dark gray), follow data from the IUCN (International Union for Conservation of Nature) (2013) and Secchi, Ott & Danilewicz (2003), respectively. Occurrences for fossil data derive from location of type localities for each taxon, except for reports for the Northern Europe by Pyenson & Hoch (2007), Western South America by Gutstein et al. (2015), and Amazonia and Eastern South America by Cozzuol (2010). Major fossil localites for enumerated inioids identified at least to the generic level, are listed alphabetically by region, and represented by teal or blue dots, for freshwater and marine deposits, respectively. Base map generated by Indiemapper (http://indiemapper.com).

Methods

Excavation at the type locality

The type specimen of this new taxon was initially discovered in an intertidal zone outcrop of the Chagres Formation, near the town of Piña, along the Caribbean coastline of Panama, in early 2011 (Fig. 2 and Fig. S1). The infrequency of low tides at the type locality created a narrow time window for excavating the specimen, which several co-authors (NDP, JVJ, DV, and AO) undertook on 18 June 2011 with the assistance of staff from Smithsonian Tropical Research Institute (STRI). After exporting the specimen under permits from Panama’s Ministerio de Comercio e Industrias (MICI number DNRM-MC-074-11) to the Smithsonian’s National Museum of Natural History (NMNH) in Washington, D.C., USA, the specimen was prepared using mechanical tools and consolidated using standard fossil vertebrate preparation techniques by DV, S Jabo, and P Kroehler in the Vertebrate Paleontology Preparation Laboratory in the Department of Paleobiology at NMNH.

Figure 2 Locality and geology.

Geographic and stratigraphic context of Isthminia panamensis. (A) Map of Central America with a yellow star representing the type locality, STRI locality 650009. (B) Map of north-central Panama with the distribution of the Chagres Formation, with type locality of Isthminia in the vicinity of Piña, along with other fossil vertebrates. (C) Chronostratigraphic and lithostratigraphic relationships of the Chagres Formation (modified from Hendy et al., in press, and Velez-Juarbe et al., 2015).

Digital methods

During excavation at the type locality (Fig. S1), we documented in situ skeletal remains using a Flip camera (Cisco Systems Inc., San Jose, California, USA) on time-lapse settings. Later, subsequent to the specimen’s preparation in the Department of Paleobiology, we used computed tomography (CT) to scan the type specimen USNM 546125 in the Department of Anthropology with a Siemens Somatom Emotion 6 at slice thickness of 0.63 mm (which results in a three-dimensional reconstruction increment of 0.30 mm). The resultant DICOM files were processed by loading image files in Mimics (Materialise NV, Leuven, Belgium), and a mask was created based on the threshold of bone, relative to the nominal density of air. We then created a three-dimensional (3D) object from this mask, and exported the resultant file as an ASCII STL, which was opened in Geomagic (ver. 2012) for final imaging edits. We also attempted to use laser surface scanning (i.e., laser arm scanner) to capture 3D data, but line of sight issues with overhanging morphological features and the geometric complexity of the type specimen prevented a full capture of the surface geometry. As a result, we elected to use the 3D models of the skull, mandibles, and scapula generated from CT data because this method provided complete capture of the external and internal morphology. After converting the CT files into 3D data, the watertight model was then processed in Autodesk Maya (ver. 2013) by Pixeldust Studios (Bethesda, Maryland, USA), decimating the models to 100,000 triangles and creating diffuse, normal, and occlusion texture maps. The resultant 3D surface model datasets, processed from the computed tomography scans, provided sub-millimeter accuracy, and full resolution files can be downloaded at the open-access Smithsonian X 3D browser (http://3d.si.edu). These files, along with supplemental ones, are also archived at Zenodo (http://zenodo.org) at the following DOI: 10.5281/zenodo.27214.

Phylogenetic analysis

Recent work on the systematics of living and extinct odontocetes has recently provided several phylogenetic frameworks to use in this study. Geisler et al. (2011) used a combined morphological and molecular analysis to clarify the relationships among extant and fossil lineages of cetaceans, with mostly a focus on odontocetes, including some important fossil taxa, but taxon sampling within Inioidea was relatively sparse compared to Geisler, Godfrey & Lambert (2012). This latter work, which described Meherinnia isoni Geisler, Godfrey & Lambert, 2012, a late Miocene inioid from marine rocks of North Carolina, USA, also included other fossil inioids such as Auroracetus bakerae Gibson & Geisler, 2009, Ischyrorhynchus vanbenedeni Ameghino, 1891, Protophocaena minima, and Stenasodelphis russellae Godfrey & Barnes, 2008, some of which were not included in subsequent phylogenetic analyses of odontocetes, such as the one by Murakami et al. (2014). The starting point for our analysis was the matrix provided by Aguirre-Fernández & Fordyce (2014) in their description of the early Miocene stem odontocete Papahu taitapu Aguirre-Fernández & Fordyce, 2014, which used the morphological partition of Geisler, Godfrey & Lambert (2012) in their description of Meherrinia, along with some important modifications (e.g., the removal of Mysticeti and unpublished specimens, and coding revisions for select stem odontocetes) that enhanced its utility for resolving fossil and living odontocete relationships.

We added Isthminia as an operational taxonomic unit to the Aguirre-Fernández & Fordyce (2014) matrix of 311 characters, and updated the character scoring for Ischyrorhynchus, which was the only inioid taxon not coded from direct observation in any previous study. The codings for Ischyrorhynchus herein were made by one of the authors of this study (CSG), who reviewed all the specimens in Argentina (e.g., MLP 5–16, MACN 15135), which resulted in modifications for 20 character codings (see File S1). The cladistic search was performed in PAUP* (Swofford, 2002) using all characters as unordered. We first performed a heuristic search using the tree bisection-reconnection (TBR) algorithm. In addition, we conducted statistical support analyses by searching for successively longer trees to calculate decay indices and 1,000 bootstrap replicates. The complete matrix is available in the Supplemental Information (see File S1).

Phylogenetic nomenclature

We followed the recommendations of Joyce, Parham & Gauthier (2004) for the conversion of select ranked taxonomic cetaceans names to phylogenetically defined ones in this study. The taxonomy of marine mammals includes several extant monospecific forms in their own familial rank, such as Eschrichtius robustus (Lilljeborg, 1861), Physeter macrocephalus Linnaeus, 1758, Pontoporia blainvillei, or Lipotes vexillifer. In many of these latter cases, the conceptual basis for the higher taxonomic rank includes many fossil taxa that connect the monospecific taxon to their nearest living relatives, especially with stem lineages that range into geologic times that remain poorly sampled and known (e.g., the Oligocene; see Uhen & Pyenson, 2007). While it would be ideal to create stem-based clade names for these single species, there remains no pathway to define pan-stems based on single species, even more than 10 years after Joyce, Parham & Gauthier (2004)’s recommendations. Here we follow Joyce, Parham & Gauthier (2004)’s logic in the specific case of the pan-stem for the lineage leading to extant Inia, by forming a new pan-stem name by combining the current Linnaean generic name with the prefix ‘pan,’ and then referred traditional family names to a more inclusive clade whose composition closely resembles our current name application. For these purposes, we used abbreviations NCN for New Clade Name and CCN for Converted Clade Name. Below, we clarify our precise definitions for these clades (see PhyloCode, 2014, Article 9.3; Cantino & de Queiroz, 2014), and we also provide full citations for the names of specifier species.

Specimens observed

Auroracetus bakerae (USNM 534002), Inia geoffrensis (USNM 395415, 49582, 239667), Ischyrorhynchus vanbenedeni (MACN 15135, MLP 5–16), Lipotes vexillifer (USNM 218293, AMNH 57333), Meherrinia isoni (CMM-V-4051, USNM 559343, identified by JA Geisler), Pontoporia blainvillei (USNM 482727, 482771, 482707), Stenasodelphis russellae (CMM-V-2234).

Nomenclatural acts

The electronic version of this article in Portable Document Format (PDF) will represent a published work according to the International Commission on Zoological Nomenclature (ICZN), and hence the new names contained in the electronic version are effectively published under that Code from the electronic edition alone. This published work and the nomenclatural acts it contains have been registered in ZooBank, the online registration system for the ICZN. The ZooBank LSIDs (Life Science Identifiers) can be resolved and the associated information viewed through any standard web browser by appending the LSID to the prefix http://zoobank.org/. The LSID for this publication is: urn:lsid:zoobank.org:pub:4763A625-883D-4263-B376-33B9F9AD56A4. The online version of this work is archived and available from the following digital repositories: PeerJ, PubMed Central and CLOCKSS.

Results

Systematic paleontology

Cetacea Brisson, 1762	
Odontoceti Flower, 1867	
Delphinida Muizon, 1988a	
Inioidea Gray, 1846sensu Muizon, 1988a	
Pan-Inia (NCN) (panstem-based version of Inia (Blainville, 1817))	
Isthminia, gen. nov. LSID: urn:lsid:zoobank.org:act:83F6A9B4-289D-45DE-A3D1-C361DAAAF973.	

Definitions ‘Pan-Inia’ refers to the panstem that includes crown Inia (CCN), and all other lineages closer to Inia than to Pontoporia, such as Isthminia and Ischyrorhynchus. Subjective synonymies of Pan-Inia include: Iniidae Gray, 1846; Iniinae Flower, 1867; Saurocetidae Ameghino, 1891; Iniidae Muizon, 1984; Ischyrorhynchinae (Cozzuol, 1996); Iniidae Cozzuol, 2010; Iniidae Gutstein, Cozzuol & Pyenson, 2014b. Crown group Inia refers to the crown clade arising from the last common ancestor of all named species of Inia, including Inia boliviensis d’Orbigny, 1834 and Inia araguaiaensis Hrbek et al., 2014. Although we follow the suggestions of the Society for Marine Mammalogy’s Committee on Taxonomy (2014) in provisionally recognizing two sub-species of Inia geoffrensis (I. g. geoffrensis and I.g. humboldtiana Pilleri & Gihr, 1977), the phylogenetic definition of Inia can accommodate a plurality of species and subspecies.

Type and only known species.Isthminia panamensis, sp. nov.

Etymology.Isthm- reflects the type specimen’s provenance from the Isthmus of Panama and the crucial role that the formation of this isthmus played in Earth history and evolution of the biota of the Americas. This epithet follows in the tradition of another fossil cetacean from the Chagres Formation, Nanokogia isthmiaVelez-Juarbe et al., 2015. The feminine generic epithet Inia reflects its similarities to the living Amazon River dolphin (Inia geoffrensis). Pronunciation: ‘Ist-min-ee-a,’ with the emphasis on the second syllable.

Age. Same as that of the species.

Diagnosis. Same as that of the species.

Isthminia panamensis sp. nov. (Figs. 3–12 and Tables 1–3)

Figure 3 Skull in dorsal, anterior, and posterior views.

Dorsal views of the type skull of Isthminia panamensis (USNM 546125) from (A) photographs and (B) orthogonal digital three-dimensional polygon model prepared from CT data, with lighting and color modifications using the Smithsonian X 3D browser. (C) Anterior and (D) posterior views of the type skull of Isthminia panamensis (USNM 546125) from orthogonal digital three-dimensional polygon model prepared from CT data. See http://3d.si.edu/explorer?s=h2mqJ9 (dorsal view), http://3d.si.edu/explorer?s=bA5gJO (posterior view), and http://3d.si.edu/explorer?s=e1seD5 (anterior view) to measure, modify, or download this model.

Table 1 Measurements of holotype skull and mandibles (USNM 546125) of Isthminia panamensis, in mm (modified after Perrin, 1975 and Tanaka & Fordyce, 2014).

	Measurement (mm)	
Skull	
Total length from the most anterior point to the posterior most point	571+	
Cranial length	185+	
Length of rostrum—from tip to line across hindmost limits of antorbital notches	381	
Width of rostrum at base—along line across hindmost limits of antorbital notches	124*	
Width of rostrum at 60 mm anterior to line across hindmost limits of antorbital notches	90*	
Width of rostrum at mid-length	36+	
Width of premaxillae at mid-length of rostrum	31+	
Width of rostrum at 3/4 length, measured from posterior end	50*	
Greatest width of premaxillae	78*	
Projection of premaxillae beyond maxillae measured from tip of rostrum to line across foremost tips of maxillae visible in dorsal view	85+	
Width of premaxillae at a line across posterior limits of antorbital notches	48*	
Maximum width of premaxillae at mid-orbit level	52*	
Preorbital width at level of frontal-lacrimal suture	184*	
Postorbital width across apices of postorbital processes	232*	
Distance from tip of rostrum to external nares (to mesial end of anterior transverse margin of right naris)	419+	
Distance from foremost end of junction between nasals to hindmost point of margin of supraoccipital crest	68	
Greatest width of external nares	49	
Median length of the nasals	58	
Maximum length of the right nasal	58	
Median length of frontals on the vertex	25	
Vertical external height of the skull from ventral most braincase to dorsal extremity of vertex	150+	
Bizygomatic width	262*	
Length of upper left tooth row—from hindmost margin of hindmost alveolus to tip of rostrum	329	
Number of teeth—upper left	18	
Number of teeth—upper right	18	
Mandible	
Maximum preserved length of left mandible	478+	
Maximum preserved height of left mandible	74+	
Number of teeth—lower left	17	
Number of teeth—lower right	18	
Length of the lower tooth row from tip of mandible to posterior margin of posterior most alveolus	315	
Notes.

Asterisk indicates doubling of measurement from one side. Positive sign indicates preserved distance.

Table 2 Measurements of the scapula (USNM 546125) of Isthminia panamensis, in mm (modified after Perrin, 1975).

Scapula	Measurement (mm)	
Maximum height of scapula	141+	
Height of scapula from posterior margin of glenoid fossa to glenovertebral angle	161	
Length of coracoid process	40	
Greatest width of coracoid process	23	
Greatest width of acromion process	26	

Table 3 Relative orbit size (ROS) in Isthminia panamensis, and in other fossil and modern odontocetes, ranked in increasing value.

Genus	Species	Specimen	ROS	Source	
Aulophyseter	morricei	LACM 154100, USNM 11230	0.20	This study (average, n = 2)	
Orycterocetus	crocodilinus	USNM 22926	0.22	This study	
Inia	geoffrensis	USNM 23967, 49582, 395415	0.24	This study (average, n = 3)	
Lipotes	vexillifer	USNM 218293	0.32	This study	
Aprixokogia	kelloggi	USNM 187015	0.34	This study	
Lophocetus	repenningi	USNM 23886	0.36	This study	
Simocetus	rayi	USNM 356517	0.36	This study	
Isthminia	panamensis	USNM 546125	0.40	This study	
Nanokogia	isthmia	UF 280000	0.40	Velez-Juarbe et al., 2015	
Xiphiacetus	bossi	USNM 8842, 175381	0.42	This study (average, n = 2)	
Delphinodon	dividum	USNM 7278	0.46	This study	
Kogia	sima	LACM 47142	0.55	This study	
Meherrinia	isoni	IRSNB M.2013	0.56	Geisler, Godfrey & Lambert, 2012	
Pontoporia	blainvillei	USNM 482707, 482717, 482771	0.57	This study (average, n = 3)	
Atocetus	nasalis	LACM 30093	0.58	Barnes, 1985a	
Kentriodon	pernix	USNM 8060	0.58	This study	
Parapontoporia	wilsoni	UCMP 83790	0.62	Barnes, 1985b	
Brachydelphis	jahuayensis	MNHN PPI 267, 268; MUSM 567, 568	0.70	Lambert & de Muizon, 2013 (average, n = 4)	
Brachydelphis	mazeasi	MNHN PPI 121, 266; MUSM 564	0.80	Lambert & de Muizon, 2013 (average, n = 3)	

LSID: urn:lsid:zoobank.org:act:A5C706B6-E0B6-43E5-A65C-E6FE0B2BDF1A

Holotype. USNM 546125, consisting of an incomplete skull, both right and left mandibles, an incomplete right scapula, and two carpals. The skull lacks the basicranium and tympanoperiotics. The holotype was collected by several of the co-authors of this study (NDP, JVJ, DV, and AO), with assistance from staff from STRI, in 2011.

Type locality. STRI locality 650009 (9°16′55.4880″N, 80°02′49.9200″W), less than 100 m northeast of the main road in the town of Piña, along the Caribbean Sea coastline of the Republic of Panama (Fig. 2).

Formation. Piña Facies of the Chagres Formation.

Age. Microfossils place the Chagres Formation in calcareous nanofossil zone NN11 and planktonic foraminiferal zones M13b-M14, suggesting an age range between 8.52 to 5.57 Ma, i.e., Tortonian to Messinian in age (Collins et al., 1996). Hendy et al. (in press) obtained a Sr date of ∼7.64 Ma from a single mollusc shell that was collected stratigraphically below the unit where the type specimen of Isthminia was discovered. This result, however, conflicts with planktonic foraminiferal data. The top of the Toro Point Member of the Chagres Formation includes co-occurring Globorotalia margaritae Bolli & Bermúdez, 1965 and G. lenguaensis Bolli, 1957. Collins et al. (1996) indicating an astronomically calibrated age (Wade et al., 2011) of 6.13–6.08 Ma. Globoquadrina dehiscens Chapman, Parr & Collins, 1934 has an age range of 23.2–5.8 Ma (Wade et al., 2011) and occurs throughout the stratigraphic section, including Panama Paleontology Project locality (PPP) 1099 (Collins et al., 1996), which is located less than 1 kilometer from STRI locality 650009, and coeval with the type specimen of Isthminia. Because the Piña Facies is stratigraphically above the Toro Member in the Chagres Formation, these observations therefore constrain the age of the type specimen of Isthminia to 6.1–5.8 Ma (Messinian).

Diagnosis.Isthminia is a medium sized crown odontocete (approximately 285 cm in total length), which can be can differentiated from all other odontocetes by the following combination of character states. First, Isthminia belongs in Inioidea based on: the presence of a very long mandibular symphysis (c. 39[2]); a fused mandibular symphysis (c. 40[0]); a lacrimal that wraps around anterior edge of supraorbital process of frontal and slightly overlies its anterior end (c. 51[1]); and the maxilla forming the dorsolateral edge of the ventral infraorbital foramen (c. 57[1]).

Isthminia is characterized by the following unique combination of character states among Inioidea: rostral constriction well anterior to antorbital notch (c. 6[1]), shared with Pontoporia; posterior edge of rostral edge bowed forming a deep U-shaped antorbital notch (c. 11[2]), shared with Brachydelphis spp.; small transverse distance between lateral edges of left and right premaxillae at antorbital notch (c. 66[0]), shared with Auroracetus and Inia; short posterolateral sulcus (c. 72[1]), shared with Protophocoena, Stenasodelphis, and Auroracetus; thickened anterolateral corner of maxilla over supraorbital process of frontal (c. 78[1]), shared with Pontoporia and Stenasodelphis; presence of a maxillary ridge (c. 79[1]), shared with Brachydelphis mazeasi; V-shaped anterior edge of nasal opening (c. 81[0]), shared with Protophocoena and Auroracetus; posterior end of premaxilla adjacent to lateral edge of nasal opening (c. 89[0]), shared with Brachydelphis spp.; suture with left and right nasals and right and left frontals shifted towards the left (c. 114[1]), shared with Pliopontos and Inia; nasals that are anteroposteriorly elongated (c. 117[0]), shared with all inioids except Ischyrorhynchus and Inia; supraoccipital below frontal and/or nasals (c. 128[0]), shared with Protophocoena, Meherrinia and Ischyrorhynchus; dorsal margin of mesethmoid at same level of premaxilla (c. 305[1]), shared with Brachydelphis mazeasi and Stenasodelphis; intermediate separation between posterior-most point of right premaxilla and nasal (c. 306[1]), shared with Pontoporia and Stenasodelphis; medial portion of maxilla on either side of the vertex face mainly dorsally (c. 307[2]), shared with Pontoporia and Pliopontos; longest side of nasal facing anterodorsally (c. 311[1]), shared with all inioids except Pontoporia (face dorsally: c. 311[0]), and Ischyrorhynchus and Inia (face anteriorly: c. 311[2]).

Among other fossil inioids in the panstem to Inia, Isthminia shares the following: with Meherrinia and Inia (not preserved in Ischyrorhynchus) three or more dorsal infraorbital foramina (c. 64[2]); with Ischyrorhynchus,  premaxillae on anterior two thirds of rostrum contact along the midline for nearly their entire length (c. 9[0]), tooth enamel with reticular striae (c. 26[1]), and anterior edge of nasals in line with posterior half of supraorbital processes (c. 80[4]); with Inia and Ischyrorhynchus supraorbital processes of frontal that slope laterodorsally away from vertex (c. 46 [2]), transverse width of nasals within 10% of nares width (c. 119[2]), nasals elevated above rostrum relative to lateral edge of maxilla (c. 123[1]), and frontals higher than nasals (c. 124[2]); with Inia the following synapomorphies: posterior buccal teeth that are nearly an equilateral triangle (c. 30 [1]), small lacrimal (c. 50[0]), small exposure of the lacrimal and jugal posterior to the antorbital notch (c. 55[0]), posterior portion of nasals elevated above rostrum (c. 123[1]), frontals posterior to nasals with same width as nasals (c. 125[1]), maxilla on dorsal surface of skull does not contact supraoccipital posteriorly (c. 129[0]), and dorsal edge of zygomatic process with distinct dorsal flange (c. 143[1]).

Lastly, Isthminia displays the following apomorphies: maxilla and premaxilla fused along most of rostrum (c. 10[0]); lower number of mandibular teeth (18) (c. 37[5]); dorsal edge of orbit low relative to lateral edge of rostrum (c. 47[1]); premaxilla is convex transversely anterior to nasal openings (c. 68[1]); posterior-most end of ascending process of premaxilla in line with posterior half of supraorbital process of frontal (c. 74[2]); very narrow width of posterior edge of nasals (c. 120[3]); slight emargination of posterior edge of zygomatic process by sternomastoid muscle fossa (c. 144[1]); and dental roots that are elongate, rugose, bulbous, and much larger than the tooth crowns, with some roots that have their apices oriented posteriorly so that they come close to the anterior end of the root of the succeeding tooth.

Etymology. The species epithet recognizes the Republic of Panama, its people, and the many generations of scientists who have studied its geological and biological histories.

Description

Skull

The skull of Isthminia is relatively complete on its dorsal aspect, although it is missing the left side of the facial bones (Fig. 3). The skull is heavily eroded along its ventral surface, and the basicranium is absent except for a small portion of the right parietal and right alisphenoid (Fig. 4). The skull preserves most of the dorsal aspect of the supraoccipital, including small portions that articulate with the vertex and nuchal and sigmoidal crests (Figs. 3A–3C). Overall, the profile of the skull is dominated by the rostrum (Fig. 5), which is complete and comprises approximately 75% of the length of the preserved skull (the rostrum length is 36.6 cm; Table 1). The anterior portion of the rostrum is slightly displaced by both an oblique and transverse fractures, likely from geologic compaction or other diagenetic factors, which displace the elements approximately 1–2 mm from their life positions. Most of the upper dentition is missing from the skull, except for the anterior teeth, some of which are complete; other more posterior teeth are incomplete, while three isolated teeth were recovered from the quarry at the type locality. Despite the heavy erosion that removed most of the left portion of this skull, sufficient anatomical details are preserved on the right side of the cranium, and along the rostrum to provide insights into the morphology of Isthminia.

Figure 4 Skull in ventral view.

Ventral views of the type skull of Isthminia panamensis (USNM 546125) from (A) photographs and (B) orthogonal digital three-dimensional polygon model prepared from CT data, with lighting and color modifications using the Smithsonian X 3D browser. See http://3d.si.edu/explorer?s=iEpExr to measure, modify, or download this model.

Premaxilla. In dorsal view, the premaxilla dominates the visible part of the rostrum, comprising the entirety of the rostrum from its anterior end to about 75% of the length of the rostrum. In this view, the premaxilla occupies a width greater than that of the maxilla until the level of the maxillary flange (sensu Mead & Fordyce, 2009:62), where the width of the premaxilla begins to taper relative to the expansion of the maxilla overlying the cranium, in dorsal view (Fig. 3). Along the rostrum, anterior of the premaxilla-maxilla suture, there are several shallow longitudinal canals that terminate in small oval foramina (∼5 mm long by ∼2 mm wide). These canals are similar to those observed in adult specimens of Inia, but markedly different from the singular, deep groove that separates the posterior connection of the premaxilla and maxilla in Pontoporia, Ischyrorhynchus, immature specimens of Inia, and Lipotes. In Isthminia, adult Inia and Lipotes, these canals disappear posteriorly, as the premaxilla-maxilla suture becomes seamless along the length of the rostrum (Figs. 3 and 5).

Figure 5 Skull in lateral view.

Right lateral views of the type skull of Isthminia panamensis (USNM 546125) from (A) photographs and (B) orthogonal digital three-dimensional polygon model prepared from CT data, with lighting and color modifications using the Smithsonian X 3D browser. See http://3d.si.edu/explorer?s=jn4ynp to measure, modify, or download this model.

The paired right and left premaxillae are unfused for 4 cm at their anterior tip (Figs. 3A, 3B and 3D), presenting a slight gap, which is likely homologous in other odontocete taxa with the mesorostral groove (sensu Mead & Fordyce, 2009:16). This gap is then obscured posteriorly by full sutural fusion between the premaxillae for 24 cm along the midline of the rostrum until an elongate (6.9 cm-long) window is exposed between the overarching right and left premaxillae, just anterior of the level of the antorbital notches (Figs. 3A and 3B). Near the anterior origin of this window, the anteromedial sulcus appears, approximately at the transverse level of the last upper tooth alveolus (Fig. 4). This latter sulcus extends subparallel to the latter window until it terminates posteriorly in the premaxillary foramen. In Inia, the anteromedial sulcus extends farther anteriorly, and the portion of the premaxilla medial to the sulcus is more bulbous, while in Pontoporia the anteromedial sulcus is deeper, and nearly enclosed dorsally by overhanging flanges of the premaxilla. Fossil pontoporiids are broadly similar to Pontoporia, whereas in Pan-Inia, such as Ischyrorhynchus and Meherrinia, this area is not well preserved. At the level of the premaxillary foramen, the right and left premaxillae diverge from their midline fusion in separate paths around the external bony naris (Fig. 6). This divergence produces a V-shaped gap, 32 mm in anteroposterior length and 9 mm in lateral width, which is narrowed and longer than fossil pontoporiids, such as Auroracetus; this gap is small and variable in Inia, and broad and triangular in Ischyrorhynchus and Meherrinia.

Figure 6 Close-up on vertex of skull.

Close-up views of the vertex in the type skull of Isthminia panamensis (USNM 546125) from (A) photographs and (B) orthogonal digital three-dimensional polygon model prepared from CT data, with lighting and color modifications using the Smithsonian X 3D browser. See http://3d.si.edu/explorer?s=cGDc1L to measure, modify, or download this model.

The premaxillary foramen itself is thinly ovate, 11 mm anteroposterior length, and 4 mm wide (Fig. 6), unlike the small, subcircular foramina in other fossil inioids. (The left side of the cranium, from this level posteriorly is not preserved, and thus the remainder of the description necessarily uses the right side of the cranium). The posterolateral sulcus is shallow, and extends slightly laterally from its deepest portion at its origin, the premaxillary foramen. The posterolateral sulcus terminates posteriorly in a faint way at the level of the anterior margin of the external naris. This condition is similar to Meherrinia and Brachydelphis, while it is different from Pontoporia, Auroracetus, Pliopontos, Pontistes and Inia, which present a deeply excavated sulcus along the posterolateral edge of the premaxilla. This portion of the premaxilla is not well preserved in Ischyrorhynchus. Medially, the posteromedial sulcus is unusual in originating 9 mm posterior of the premaxillary foramen and bifurcating into lateral and medial tracts that delineate the borders of the premaxillary sac fossa. Along with the posterolateral sulcus, these bifurcating tracts create a Z-shaped sulci pattern that is shallow laterally and deep (>3 mm) anteromedially (Fig. 6B). The path of medial tract of the posteromedial sulcus extends along the lateral margin of the anterior half of the external naris, but it is not confluent with the border of the naris. This morphology is completely new, and not observed in any inioid nor delphinidan. The bifurcating tracts enclose a low, but convex premaxillary sac fossa located lateral to the external naris and dipping medially, whereas the premaxillary sac fossa in all other inioids is located anterolateral of the external naris and is strongly convex, except for Meherrinia and Pliopontos. This portion is not preserved in Ischyrorhynchus. The premaxillary sac fossa in Lipotes is flat, with elevated margins.

The patent posterior termination of the entire premaxilla is spatulate, flat, and it appears at the level of the posterior half of the external bony naris, as in Meherrinia. There is an 8 mm separation between the posteomedial termination of the premaxilla and the anterolateral-most point of the nasal. In contrast, the posterior termination of the premaxillae of Pontoporia reaches the level of the posterior edge of the external nares, while in adult Brachydelphis spp., Pliopontos, Pontistes, Inia, and Lipotes, it extends even farther posteriorly; in young specimens of Brachydelphis and Pontoporia, it is in an intermediate position. Although there is slight erosion of the bony surface along the immediate margin of the external naris, the gap between the premaxilla and nasal is patent.

Maxilla. Throughout most of the anterior two thirds of the rostrum, the maxillae and premaxillae have a cylindrical outline (Fig. 3). Dorsally, the maxilla is exposed slightly on the lateral margin of the rostrum that is otherwise dominated by the premaxilla until about the proximal third of the rostrum where the maxilla becomes flatter along the maxillary flange. (As with the premaxilla, nearly all of the facial portion of the left maxilla has been lost to erosion, and the description is based on the right side). The antorbital notch is widely open, U-shaped, and oriented anteriorly. Posterior to the antorbital notch, the maxilla is expanded to cover most of the supraorbital process of the frontal, with the exception of the posterior-most and posteromedial edge, where the frontal is exposed. This posteromedial exposure of the frontal is similar to the condition observed in Ischyrorhynchus and Inia (mainly in juveniles), and differs from Pontoporia, Pontistes, Pliopontos, Meherrinia, Brachydelphis spp., and Lipotes, where the maxillae reaches the nuchal crest, and the lateral edges of the vertex. Posterolateral to the antorbital notch, the maxilla form a low maxillary crest (sensu Mead & Fordyce, 2009:51), which extends from the preorbital process, continues along the length of the supraorbital process of the frontal, but terminates at the postorbital process, unlike in Inia, where the crest continues well posterior of the postorbital process and join the temporal crest. In Isthminia, the maxillary crest is mediolaterally thicker (2–6 mm), but lower (∼5 mm), than the thinner, but higher (>5 mm) crest observed in Inia; in Pontoporia and Pliopontos this crest extends only the length of the supraorbital process.

Dorsally, the right maxilla shows a large diameter (∼10 mm) anterior dorsal infraorbital foramen, located at the level of the antorbital notch (Figs. 3A, 3B, 3D and 6). A second, anterior dorsal infraorbital foramen is posterolateral to the first one, and it is smaller in diameter (∼7 mm), and oriented posterolaterally. A single, posterior dorsal infraorbital foramen is located posterolateral to the external nares, it has a diameter of about 9 mm and its orientation is posterodorsal. The posterior dorsal infraorbital foramen of Isthminia is absolutely larger and located farther posteriorly than the corresponding foramen in Inia, Ischyrorhynchus, Meherrinia, Brachydelphis, Pontistes, Pliopontos, Pontoporia, and Lipotes.

In ventral view, the rostral portion of the maxilla bears alveoli for at least 14 maxillary teeth, with thin interalveolar septa (Fig. 4). At the ventral midline contact between the maxillae, there is a longitudinal groove that extends from anteriorly to about the level of the fifth maxillary tooth; a similar sulcus is also observed in Inia and Pontoporia, whereas this groove reveals a palatal exposure of premaxilla and/or vomer in Ischyrorhynchus and Brachydelphis mazeasi. Along the ventral surface and anteromedial to the jugal, there is a shallow (∼2 mm) oval (∼17 mm long by 10 mm wide) fossa; a similar fossa is also present in some specimens of Inia, Ischyrorhynchus, Brachydelphis spp. and very slightly Pontoporia. Medial to this shallow fossa, which we term the ventral maxillary fossa, there is an elongated fossa that continues anteriorly parasagittally for about 60 mm, and 5 mm in width and depth. The location and morphology of this latter fossa corresponds to the anterior sinus of Inia (Fraser & Purves, 1960), and it is exposed in Isthminia because its overlying maxilla and palatine were eroded. An anterior sinus is also found in Ischyrorynchus, however it is shorter than that in Inia and Isthminia. The rostral portion is not preserved in the other genera of inioids, preventing any comparison.

Lacrimal and Jugal. The lacrimal appears to be ankylosed with the anterior margin on the supraorbital process of the frontal, forming its anterior surface, a condition common to all adult inioid specimens (Figs. 3–5). Ventrally, the lacrimal extends medially to join the jugal, which together forms the anteroventral surface of the antorbital notch. The preserved part of the jugal is a thin strut that is subcylindrical in outline (∼4 mm wide; 17 mm long; ∼2 mm thick) and oriented posteroventrally. Overall, it is very similar in morphology to the jugal of Inia.

Frontal. Dorsally, the frontal is mostly covered by the maxilla, but it is exposed along the posterior and posteromedial edges of the skull roof (Figs. 3 and 5–7). In Isthminia, the right and left frontals form the highest part of the vertex, and together form a pair of rounded, rectangular knobs with a slight midline cleft (Figs. 3A–3C, 5 and 6). This topographic high for the frontals at the vertex is similar in Inia, Ischyrorhynchus or Meherrinia, and even Pontoporia and Lipotes, although the frontals in Isthminia are small and low by comparison with Pan-Inia. Unlike Inia and Meherrinia, the midline cleft between the right and the left frontals at the vertex does not show participation of an anterior supraoccipital (or possibly interparietal) wedge externally nor in internal CT scan data (Fig. 7 and Video S1). The dorsal surface of the vertex is lightly rugose, but not as strongly as in adult specimens of Inia.

Figure 7 Transverse CT slices through the skull.

Computed tomography (CT) slices through the vertex of Isthminia panamensis (USNM 546125) across four slightly sub-transverse planes that pass anterior to posterior (A–D). CT slices (A–D) represent respective CT slices numbers 20566, 20625, 20655, and 20708, available for download on the Smithsonian X 3D browser (http://3d.si.edu). Numbers 1 and 2 denote facial and endocranial sagittal midlines, respectively, showing the sinistral displacement of the facial bones typical in many odontocetes (Geisler & Sanders, 2003; Mead & Fordyce, 2009).

The supraorbital process is dorsoventrally thin (∼5 mm) with a blunt preorbital process; in contrast, the postorbital process is more elongated with a triangular cross section through its longitudinal axis, similar to the general condition of the other inioids. Nevertheless the distance between this two processes (52 mm), reflecting the size of the orbit, is about twice that of adult specimens of Inia, but in Isthminia it is proportionally similar to the other fossil inioids (all known specimens of Ischyrorhynchus lack this feature; see Table 3). In dorsal view, the lateral edge of the supraorbital process is relatively straight and oriented parasagitally, unlike Inia and Pontoporia where this border is laterally concave and oriented anterolaterally, or the nearly straight but anterolaterally oriented borders of Pliopontos and Brachydelphis. Additionally, the postorbital process is shorter than the length of the orbit, contrasting with the much longer process and smaller orbit in Inia. The ventral surface of the supraorbital processes is gently concave with a low, but distinct postorbital ridge. Medially and posterior to the frontal groove there is a shallow (<1 cm) round (∼1.5 cm diameter) fossa for the postorbital lobe of the pterygoid sinus. This same fossa varies tremendously in adult specimens of Inia, where it can either be shallow and slit-like (e.g., USNM 49582) or form a deep pit (e.g., USNM 239667). By contrast, this fossa in Pontoporia is deep, rounded and floored posteroventrally by the alisphenoid; in Brachydelphis spp., this fossa is shallow, as it is in Lipotes.

In the ventral view of the endocranium (Fig. 4), the right and left frontals surround the anterior aspect of the endocranium, where the extensive cerebellar juga are preserved on the ventral surface (Mead & Fordyce, 2009:18). Medially, the posteromedial margins of the frontals inside the endocranial region enclose a deep dorsal sagittal sinus sulcus along the midline. Such a structure is not visible in intact, extant skulls of Inia and Pontoporia, available for observation, nor is it preserved in most fossil inioids. Incomplete crania of Brachydelphis referable to B. jahuayensis (Gutstein et al., 2009: Fig. 7B) show no such sinus, but instead a low, bony ridge. Finally, a small wedge of the supraoccipital directly ventral to the vertex separates the fan-like posterior-most margins of the right and left frontals, which eventually contact the anterodorsal margins of the parietals along the frontoparietal sutures.

Nasal. The right and left nasals are paired at the vertex, sloping away from the topographic high of the paired frontals (Figs. 3, 5 and 6). Overall, the nasal is large (width = ∼12 mm; length = 41 mm), dominating the anterodorsal surface of the vertex, and occupying the entire posterodorsal margin of the external bony naris. The anterior margin of nasal is concave. Together, the right and left nasals are anteroposteriorly elongate with some tapering posteriorly, as in Pontoporia, Brachydelphis, Pontistes, Auroracetus, Pliopontos. However, the nasal in Isthminia is dorsoventrally more massive than these latter genera, and it is not as anterodorsally inclined as in Meherrinia not as anterior-facing as in Ischyrorhynchus, Inia, and Lipotes.

The anterior margin of the nasal displays a low sigmoidal crest that extends transversely with a small protuberance that rises in the middle of the nasal, about 10 mm from its anterior margin; with the paired right and left nasals, these small crests and the base of these protuberances outline a wide, but shallow V-shaped concavity, pointing posteriorly (Figs. 3A, 3B and 3D). The posterior margin of the nasal is difficult to resolve without close inspection because the sutural distinction between the nasal and the frontal in this part of the vertex is overlapping and thin (see also Fig. 6). The posterior termination of the nasal overlaps with the frontal by passing in a broadly posteromedial path, terminating anterior of the level of the posteriormost margin of the maxilla. Together, the posterior termination of the right and left nasals show an anteriorly-pointed V-shaped margin. This condition is similar to Pontoporia and Brachydelphis, where the contact between the nasal and frontal shows a similar V-shaped margin; in Auroracetus and Meherrinia, a small wedge of the frontals insert medially between the nasals.

Vomer and Ethmoid. The vomer is poorly preserved ventrally, but a small portion is patent along the midline palatal surface adjacent to the medial margin of the highly eroded right maxilla, approximately extending 45 mm, with an anterior extent to the transverse level of the 8th maxillary tooth alveolus (Fig. 4). The ethmoid is incompletely preserved; the crista galli is shallow with very small (<1 mm) foramina in its surface. The ethmoid forms the bony nasal septum, rising dorsally to the same horizontal level as the premaxillae, but not quite reaching the level of the nasals. The lateral wings form the posterior and posterolateral walls of the external nares, which are cleanly separated from the anterior margin of the nasals by a continuous gap 5–8 mm wide (Fig. 6).

Parietal. The parietal is exposed broadly on the posterior margin of the temporal fossa, along with the frontal and squamosal (Figs. 3C, 3D, 5 and 7). The lateral surface of the parietal is smooth and convex; in posterior view, the temporal crest of the parietal is posterolaterally oriented, as opposed to the ventrally oriented crests in Inia and Pontoporia. The anterior extent of the parietal is unclear because the parieto-frontal suture is not patent, similar to adult specimens of Inia.

Supraoccipital. Only the dorsal half of the supraoccipital can be reliably determined for Isthminia. Dorsally, the supraoccipital does not participate in the vertex, but participates in the temporal and nuchal crests (Figs. 3A–3C); the nuchal crest is transversely straight, about 10 mm thick, and unlike the more anteromedially oriented crest in Inia and the posteriorly concave crest of Pontoporia. Posteriorly, there is a midline external occipital crest that is bounded laterally by deep (9 mm) semilunar fossae; such fossae are also patent in adult specimens of Inia and Pontoporia. The external surface is smooth and convex. The temporal crests are nearly vertical, and dorsally they join the nuchal and orbitotemporal crests (sensu Fordyce, 2002:194), forming a tabular, triangular surface at the triple junction. When viewed posteriorly, the supraoccipital has a square outline, unlike the more sub-triangular outline in Inia, or the general pentagonal outlines of Pontoporia and Lipotes.

Squamosal. The right squamosal is nearly completely preserved. The zygomatic process of the squamosal is relatively long, mediolaterally thin, laterally convex, and medially concave. Overall, its main corpus is rectilineal in lateral view, in contrast to the gently tapering profile of Pontoporia and acute tapering in Inia. In Isthminia, the anterior tip of the zygomatic process is expanded, with a squared-off anterior margin, more like Inia, and to a lesser degree Brachydelphis mazeasi, rather than the rounded, tapering tip of Pontoporia and Pliopontos. The dorsal surface of the root of the zygomatic process in Isthminia is concave, while its lateral edge flares outward about 10 mm farther laterally than the anterior part of the process (Figs. 3–5). In lateral and ventrolateral views, the postglenoid process is not patent, but it shows no indication of supporting elaboration, such as the bulbous postglenoid process in both Inia and Pontoporia, and acute and thin in Brachydelphis spp. (Gutstein et al., 2009; Lambert & de Muizon, 2013). Ventrally, the outline of the glenoid fossa in Isthminia is elongate, shallowly convex, and faces ventromedially. The tympanosquamosal recess extends as a deep (∼5 mm) sulcus medial to the glenoid fossa, as it does in other inioids. The posterolateral surface of the squamosal has a broad and relatively deep concave sternomastoid fossa, deeper than Inia.

The squamosal plate is relatively low, occupying only about the lower quarter of the surface of the temporal fossa, which is dominated by the parietal (Fig. 5). This configuration is similar to the condition seen in Pontoporia and Brachydelphis, but contrasts with Inia, where the squamous portion is much higher, a condition also visible in Lipotes. The anterior extent of the squamosal plate is ankylosed with the posteroventral edge of the temporal wall exposure of the alisphenoid in the type specimen of Isthminia.

Alisphenoid. Only the dorsal portion of the alisphenoid is preserved in the type specimen of Isthminia above the horizontal level the squamosal fossa (Fig. 5). In lateral view, the parieto-alisphenoid suture extends in a path from the squamosal plate at the posterior margin of the temporal fossa dorsally to a level in line with the nuchal crests; in this way, this sigmoidal suture partitions the parietal (dorsally) and the alisphenoid (ventrally) in the middle of the temporal fossa. The anterior margin of the alisphenoid extends at least to the level of the postorbital processes of the frontal, although the actual sutures are not patent at the anterior end (see also Fig. 7). In lateral view, the dorsal extent of the alisphenoid on the temporal wall is much greater than that seen in Inia, but we note a degree of variability in Inia.

Mandible

Both right and left mandibles are preserved intact and remain articulated via an osseous symphyseal articulation (Figs. 8 and 9; Class IV jaw joint of Scapino, 1981). The length of the mandibular symphysis (21.0 cm) is approximately 43% of the entire length of the mandible. The mandibles possess nearly all of the original lower teeth; the lower first incisors are missing, along with posterior most three teeth of the right mandible (although one isolated tooth is a perfect fit for PC12; see Fig. 10). Both the right and left mandibles possessed 18 and 17 lower teeth, respectively, although the degree of bone remodeling posterior of left PC13 leads us to presume that 18 teeth is the likely maximum lower tooth count (Fig. 10). Posterior margins are incomplete for both sides of the mandible, and the left angular process appears intact and there is a weak suggestion of the osteological structure where the left articular condyle would have been. The right articular condyle is missing. Most of the mandibles are well preserved, although much of the right acoustic window is degraded from erosion and/or diagenesis (Fig. 9).

Figure 8 Mandibles in dorsal, anterior, and posterior views.

Dorsal views of the mandibles of Isthminia panamensis (USNM 546125) from (A) photographs and (B) orthogonal digital three-dimensional polygon model prepared from CT data, with lighting and color modifications using the Smithsonian X 3D browser. (C) Anterior and (D) posterior views of the mandibles of Isthminia panamensis (USNM 546125) from orthogonal digital three-dimensional polygon model prepared from CT data. See http://3d.si.edu/explorer?s=hhl3iu (dorsal view), http://3d.si.edu/explorer?s=cgvhM3 (posterior view), and http://3d.si.edu/explorer?s=gR4Rhv (anterior view) to measure, modify, or download this model. Parentheses denote missing structure(s).

Figure 9 Mandibles in ventral and lateral views.

Ventral views of the mandibles of Isthminia panamensis (USNM 546125) from (A) photographs and (B) orthogonal digital three-dimensional polygon model prepared from CT data, with lighting and color modifications using the Smithsonian X 3D browser. (C) Left lateral and (D) right lateral views of the mandibles of Isthminia panamensis (USNM 546125) from orthogonal digital three-dimensional polygon model prepared from CT data. See http://3d.si.edu/explorer?s=cavfn3 (ventral view), http://3d.si.edu/explorer?s=dGTRVj (left lateral), and http://3d.si.edu/explorer?s=cLO5aZ (right lateral) to measure, modify, or download this model. Parentheses denote missing structure(s).

In anterior view and posterior views (Figs. 8C and 8D), the mandibles show slight asymmetry in the relative directions of the overall mandibular rami, with the right ramus extending laterally and slightly ventral relative to the left one. This asymmetry may be diagenetic and related to sediment compaction, but we think it more likely records the original right-left asymmetry that is common in other living inioids (Werth, 2006), and this condition is evident in adult specimens of Pontoporia, with its proportionally elongate rostrum. In ventral view, the anterior termination of the mandibles from the gnathion to pognion is gradual and not acute, with a ventral outline that is somewhat rectangular. Anteriorly, this termination is flat and not acute. Posteriorly, the ventral surface of the mandibles is U-shaped, in transverse section, through the symphysis. Generally, this morphology is most similar to that of Inia, and Saurocetes argentinus Burmeister, 1871, which is only known from a mandibular fragment that is less complete than Isthminia (Cozzuol, 1989; Cozzuol, 2010). The general lateral and horizontal profiles of the mandible in Isthminia are unlike Pontoporia, with a deep lateral groove, and unlike the strongly convex mandibles of Brachydelphis mazeasi (based on MUSM 887).

The ventral margins of the mandible, posterior of the symphysis, are rounded until the posterior half of the level of the acoustic window when this margin gradually gains an edge (Fig. 8D). The medial profile of the acoustic window in Isthminia is dorsoventrally narrower than that of Inia, and considerably more acute than Pontoporia. Both right and left mandibles show approximately 7 mental foramina each, spaced along the ventrolateral margins of the mandibles along the symphysis. In each case, the foramina open anteriorly, often forming sulci with long tails. The anterior most foramina are paired close to the midline of the symphysis at the level in between the third and fourth lower tooth. Isthminia shares a high number of mental foramina with Inia, whereas both Pontoporia and Brachydelphis mazeasi shows fewer (1–2 mental foramina in adult specimens of Pontoporia, and 4 mental foramina in MUSM 887).

The overall morphology of the mandibles in Isthminia shares the most similarities with Inia, among inioids and delphindans for which this element is known, especially in lateral and horizontal profiles anterior to the symphysis. Posterior of the symphysis, the rami of the mandibles are lower than Inia, and slightly more gracile. The mandibles of Isthminia are also not dorsoventrally flattened like those of Pomatodelphis inequalis Allen, 1921, nor are they slender like those of Kentriodon pernix Kellogg, 1927 (USNM 8060) and Brachydelphis mazeasi (based on MUSM 887). The mandibles of Isthminia differ strongly from Lipotes, and fossil delphindans such as Lophocetus pappus Kellogg, 1955 (USNM 15985) and Hadrodelphis calvertense Kellogg, 1966 (USNM 23408, 189423), which all notably have many more teeth posterior of the symphysis, and exhibit rounded, nearly circular alveoli. Ovate alveoli are notable in putative inioids represented by fragmentary mandibles, such as Saurocetes argentinus and Hesperocetus californicus True, 1912 (UCMP 1352), although the dentition of Isthminia is far less bulbous than either. In Goniodelphis hudsoni, another putative inioid, the mandibles are relatively deeper, and mediolaterally flattened, with a much longer symphysis, and mediolaterally flattened teeth that are triangular in outline when viewed laterally, and with crowns are much more slender and somewhat recurved (see below).

Dentition

Upper. The upper dentition consists of 15 teeth per side, counted by alveoli in the premaxilla and maxilla on the right side of the skull. It is less complete than the lower dentition. Of the original upper dentition, only a total of 14 teeth remain preserved in their alveoli, with 6 in the left side and 8 in the right. Of these intact teeth, the right side preserves only the 2 distalmost teeth with crowns, while the others only preserve the tooth roots, with fractures at the base of the crown that are probably postmortem. An isolated upper right tooth discovered during excavation fits well in the third postcanine (PC3) alveolus, and the lack of any preserved alveoli posterior to this level increases the likelihood of this placement being correct, although there is no way to eliminate a more posterior placement (see Fig. 10). Another isolated tooth root lacking the crown likely belongs to a right alveolus in the posteriormost dentition that is not preserved on this side of the skull. The left side preserves intact teeth, with crowns, from the first incisor (I1) to PC1 and then an open alveolus at PC2, followed by two tooth roots with rounded breaks where crowns were likely present prior to death. Right PC7 is intact, although all of the other alveoli on this side are missing their teeth.

Figure 10 Close-up of upper and lower dentition.

The dentition of Isthminia panamensis (USNM 546125) in close view. (A–E) Upper dentition including the rostrum (A) and isolated teeth collected near the skull at the outcrop surface, showing (B), an upper left posterior tooth (likely PC3) and (C), an upper left posterior tooth. (F–I) Lower dentition including the mandible (F, G), shown in two parts, with overlapping images over the mandibular symphysis. (H–I) An additional isolated left tooth posterior (almost certainly PC12) was collected at the type locality. Dashed lines with arrowheads indicate alignment for the occlusion of upper and lower dentition.

Overall, the teeth have slightly anteroposteriorly expanded tooth roots, exhibiting an ovate outline in occlusal profile at the margin of the alveolus, which is very similar to Goniodelphis, Hesperocetus and Ischyrorhynchus, although Isthminia has more clearly ovate tooth alveoli than all of these. By comparison, Inia and Lipotes have subcircular tooth outlines at the alveolar margins, whereas Pontoporia show nearly rectangular outlines. The posterior roots of the upper teeth are somewhat gibbous, with closed pulp cavities distally. The exposed base of the tooth roots, ventral of the level of alveolar margin, tapers dramatically towards the base of the tooth crown, with the crown situated more or less centrally on the tooth root, except for the anteriormost pairs of incisors, which are slightly procumbent. The base of the upper tooth crowns range from 11–12 mm in diameter, with very light longitudinal striae that surround the perimeter of the base (such light striations are visible on both lower and upper teeth). The enamelocementum boundary between the roots and the crown is distinct and sharp for both upper and lower teeth. The apices of the upper tooth crowns are worn, leaving subcircular tooth wear outlines through the enamel into the dentin that is polished. With the exception of the first incisors, the crowns of the upper dentition exhibit a slight buccal curve. Wear facets can be noted on the posterior margins at the base of the tooth crown in the first incisors and on the anterior side of right PC1.

Lower. The lower dentition is nearly complete, consisting at most of 18 teeth per side, and missing only the first lower incisors and the two posteriormost left postcanine teeth. The right side consists of 18 teeth, whereas the left side consists of 17 teeth, although there are signs of bone remodeling where the alveolus of PC14 may have been. An isolated lower left tooth found during discovery quarrying fits reasonably well in the left PC12 alveolus, and the morphology and wear on the tooth crown matches its intact right counterpart (see Fig. 10). Like the upper dentition, the lower teeth posterior of the incisors are broadly ovate in occlusal profile, formed by the margins of the alveoli.

The near complete lower dentition provides detailed information about the morphology of the tooth crowns throughout the mandible for which the upper dentition only provides limited information. While the lateral profile of the lower dentition shows that the teeth are generally oriented vertically, but viewed along the major axis of the mandible, the anterior teeth from the canine (C1) to PC3 show buccal curvatures with slight lateral compression and mesiodistal keels that grade into straighter teeth without mesiodistal keels posterior of PC3 and that also have more apical tooth wear, leaving less of the original tooth crowns. Generally, lower dentition posterior of PC3 are rounder in occlusal profile, with slight lingual protuberances on the crown beginning at PC6 that become more patent as true lingual cusps posterior of PC9. After this level, the lower teeth grade slowly to presenting a more lingual orientation. Posterior of the termination of the mandibular symphysis, the diastemata shorten between adjacent lower teeth, although there is still enough space between the posterior most teeth to permit interlocking occlusion with the corresponding upper dentition. Most of the lower teeth lack non-occlusal wear facets, except for the right I2 and left PC9.

Careful manual articulation of the lower jaw with the rostrum using full size 3D prints of the type specimen shows that the lower and upper dentition interlock in a precise, alternating way similar to extant odontocetes (e.g., Tursiops Gervais, 1855) with robust dentition. Although both lower teeth and upper teeth have crown base diameters in the same range (11–12 mm in mesiodistal diameter), the slightly shorter lower dentition diastemata provides the space for upper and lower teeth to slide past one another. Unusually, I2–3 together pass posterior and anterior of I1–2, respectively, although such imprecise occlusions do occur in other odontocetes, and such a similar pairing in the dentition can be observed in Inia (the posterior lower teeth of USNM 49582).

Scapula

Only the right scapula is preserved in the type specimen of Isthminia (Fig. 11). In dorsoventral dimensions, the preserved element is 16.8 cm tall, and approximately 15 cm in anteroposterior length (Table 2). The scapula is incomplete, and the following parts are missing from the type specimen: most of the dorsal margin, and especially most of the anterior aspect; most of the acromion; and the anterior tip of the coracoid process. The posterior margin of the suprascapular border is intact, as well as the glenoid fossa and most of the region surrounding the ventral aspect of the scapula.

Figure 11 Scapula in lateral, medial, and distal views.

Right scapula from Isthminia panamensis (USNM 546125) in lateral (A–B), medial (C–D), and distal (E–F) views. Each respective paired view shows photographs alongside orthogonal digital three-dimensional polygon model prepared from CT data, with lighting and color modifications using the Smithsonian X 3D browser. See http://3d.si.edu/explorer?s=dmsTMl (lateral view), http://3d.si.edu/explorer?s=jPwTGO (medial view), and http://3d.si.edu/explorer?s=hwGm9I (distal view). Anatomical terminology for the scapula follows Tanaka & Fordyce (2015) and Uhen (2004).

The scapula is broadly fan-shaped, although it is exceedingly thin along the broken dorsal border, ranging from 1–3 mm in mediolateral thickness (Fig. 11). Nearly the entire part of the scapula housing the supraspinous fossa is missing, and only the basal 2 cm of the spinous process at its L-junction with the base of the acromion is preserved. The infraspinous fossa is deep, and it is the most concave aspect of the scapular topography in lateral view. Consequently, in medial view, the costal surface of the scapula shows corresponding and marked convexity. The depression for the teres major muscle is shallow, but patent. In dorsal view, the most striking aspect of the scapular morphology is the sinusoidal profile of the dorsal border created by the deep infraspinous fossa.

The acromion is incomplete, but the preserved base shows that it was dorsoventrally tall (25 mm) relative to the same dimension of the coracoid process, thin (4 mm in mediolateral thickness), and curved medially from its base; reminiscent of the condition observed in Inia. This morphology differs from the anteriorly rounded, subtriangular outline of the acromion of Brachydelphis mazeasi (MUSM 887) and Pontoporia, where the proximal end of the acromion is dorsoventrally broad and tapers distally. In lateral view, the angle formed by the acromion and the spinous process in Isthminia is nearly 90°, and the anterior margin of the scapular border bisects this angle at about 70°from the dorsal margin of the acromion. The coracoid is stepped medially from the level of the acromion, and it is thicker laterally than the acromion, with a slight lateral curve, and presents a slightly spatulate anterior termination, which is typical in delphinidans.

The glenoid fossa is 13 mm deep at its deepest, relative to its ventral margins. In ventral view, the overall shape of the glenoid fossa is roughly that of a slightly laterally compressed oval (Figs. 11E–11F); when combined with its depth, the overall topography of the glenoid fossa is reminiscent of an ice cream scoop. A sharp posterior margin of the posterior scapular border extends to the margin of the glenoid fossa.

Carpals

Two carpal elements were collected in close proximity to the cranial elements of Isthminia, disarticulated and in isolation (Fig. 12). Both elements are mediolaterally flattened with anterior, posterior, proximal, and distal surfaces that are shallowly concave to convex, forming articular surfaces with the radius, ulna, other carpals, or metacarpals. Both carpal elements also have one surface that is well preserved, while another that is highly eroded. It is difficult to side isolated cetacean carpal elements; the only other preserved postcranial element is the right scapula, which provides one argument for considering these isolated elements as belonging to the right side, although we cannot exclude the possibility that they each belong to different sides.

Figure 12 Carpal elements.

(A) Complete, intact left pectoral limb of Inia geoffrensis (USNM 395602), showing all of the individual osteological elements in articulation. Carpal elements belonging to Isthminia panamensis (USNM 546125) include (B) a possible pisiform; and (C) a likely unciform, with (D) a close up of the carpal bones in (A), for comparison.

We compared these two isolated carpals with an articulated forelimb of Inia geoffrensis (USNM 395602, see Fig. 12) as well as with other odontocetes (Cooper et al., 2007). The smallest carpal element (Fig. 12B) represents either the cuneiform or the pisiform. It has a roughly lozenge outline, and it is about 50% smaller than the larger carpal element, which makes its identity as the pisiform more likely, given its association with the larger carpal. Interpreted as a pisiform, its anterior and distal surfaces are flat and likely articulated with the cuneiform and metacarpal V, respectively. A small pisiform is observed in Inia, as well as in other delphinoids (Cooper et al., 2007), while it seems to be lost in other, more distantly related river-inhabiting taxa (e.g., Platanista gangetica, USNM 172409). The larger carpal (Fig. 12C) has an irregular pentagonal outline, which limits its identity to the unciform, cuneiform, or lunate. The proximal facets of the unciform and cuneiform articulate with metacarpals IV and V, respectively. Given the length of the longest articular facet of this element, in direct comparison with the articulated forelimb of Inia (USNM 395602), we propose that this element most likely corresponds to the unciform.

Phylogenetic analysis

We obtained six most parsimonious trees (length = 1,922; ensemble consistency index = 0.283, and ensemble retention index = 0.451), in our phylogenetic analysis, with the strict consensus cladogram shown in Fig. 13. The resulting topology is overall very similar to that obtained by Aguirre-Fernández & Fordyce (2014) (see their Fig. 8), with the notable difference that the relationship of Pontoporia, Brachydelphis and Pliopontos with other inioids which is unresolved in our analysis, yielding a polytomy for Pontoporiidae (sensu Geisler, Godfrey & Lambert, 2012). Our results also resolved a clade (Pan-Inia) of taxa more related to Inia than Pontoporia, which consists of: Meherrinia, Ischyrorhynchus and Isthminia, the latter which is sister to Inia. Although Bremer support values for most of these nodes is low (i.e., 1 step), there is stronger support (i.e., 2 steps) for the clade that includes Ischyrorhynchus + Isthminia + Inia. The new position of Ischyrorhynchus is likely a result of our rescoring of several characters based on observations of the type and additional specimens of Ischyrorhynchus. This position differs from all previous phylogenetic analyses (e.g., Geisler, Godfrey & Lambert, 2012; Aguirre-Fernández & Fordyce, 2014) but it is consistent with Cozzuol (2010)’s proposal for a subfamily grouping of Ischyrorhynchinae within Iniidae (Cozzuol, 1996). Our analysis did not include Saurocetes spp., a large Pan-Inia known from the late Miocene age Ituzaingó Formation of Argentina and Solimões Formation of Brazil, and represented mainly by fragmentary mandibular remains (Cozzuol, 1996; Cozzuol, 2010). We also did not include Goniodelphis hudsoni from the Mio-Pliocene age Bone Valley Formation of Florida (Allen, 1941), which is represented by a poorly preserved cranium with some similarities to Ischyrorhynchus. Both taxa require reexamination that remains outside the scope of this study.

Figure 13 Strict consensus cladogram.

Phylogenetic analysis of Isthminia and other inioid odontocetes, showing a strict consensus cladogram resulting from six most parsimonious trees, 95 steps long, with the ensemble consistency index equal to 0.283 and the ensemble retention index equal to 0.451. Numbers below nodes indicate decay index/bootstrap values; stem-based clades are indicated by arcs, while open circles denote node-based clades.

Our results differ in resolving a clade grouping Lipotes, Platanista and the fossil lipotid Parapontoporia spp., which shares some similarities with Platanistoidea sensu Simpson (1945) and Geisler & Sanders (2003). The recovery of Platanista in a close relationship with other Lipotes has previously been recovered in the exclusively morphological analyses of Geisler & Sanders (2003) and Aguirre-Fernández & Fordyce (2014), whereas exclusively molecular and combined molecular and morphological analyses consistently recover Platanista as a separate, basal branching clade from Lipotes and Inioidea, likely reflecting long branch attraction (see Geisler et al., 2011: Figs. 1 and 2, and references therein). Regardless, both morphological and molecular (and combined) analyses have consistently recovered Inioidea as a clade (i.e., Inia and Pontoporia), a finding replicated by our own results, herein.

Discussion

Isthminia compared with other living and extinct inioids

Among inioids, the general morphology of Isthminia in dorsal view most resembles the known elements of Meherrinia and Inia, although the broad circular outline of the maxillae and their contact with the vertex is also reminscient of Brachydelphis. In ventral view, Isthminia is most similar to Ischyrorhynchus and Goniodelphis, although both of these taxa are represented by more fragmentary remains than Isthminia. The rostrum of Isthminia is robust, with dorsal fusion between the right and left premaxillae, and possessing relatively robust upper and lower dentition, with strong wear on the apical crowns, although Isthminia does not exhibit lingual cusps in the posterior dentition observed in Inia. Additionally, tooth counts are more similar to Inia, certainly more so than Pontoporia. The strong groove separating the premaxilla and maxilla along the length of the rostrum is most similar to Inia, whereas Pontoporia and Ischyrorhynchus show a small but deep indentation that runs the length of the rostrum. In some ways, the rostrum of Isthminia is reminiscent of Kampholophos serrulus Rensberger, 1969 (UCMP 36045), from the late Miocene of California, which has dentition that is far more crenulated than Isthminia. In several basic traits (e.g., robust dentition reduced in number, robust rostrum, and a broad exposure of the temporal fossa), Kampholophos shares many similarities with Pan-Inia, although its phylogenetic position has not been determined beyond potential membership in Delphinida (see Salinas-Márquez et al., 2014).

Isthminia exhibits a large dorsal infraorbital foramen on the maxilla, which is proportionally similar to Inia and Ischyrorhynchus, although absolutely larger in Isthminia (Figs. 3 and 6). In ventral view, Isthminia shows anteriorly elongate anterior sinus system, invading the maxilla, a feature observed also in Inia (Fraser & Purves, 1960). Overall, the lateral profile of the rostrum in Isthminia remains in the same level as the cranium, whereas both Pontoporia and Inia shows a slightly dorsal elevation of its orbits, a featured most pronounced among odontocetes in Lipotes. Using the small crest on the supraoccipital as an external demarcation of the hemispherical midline of the underlying dermocranium, we note that the vertex in Isthminia is slightly sinistral (see also Fig. 7), to the same degree as Inia, and more so than Pontoporia, although not as highly sinistral as Lipotes. Interestingly, Isthminia lacks the strongly elevated and knob-like vertex of Inia and Ischyrorhynchus, maintaining a lower vertex profile similar to Meherrinia, Brachydelphis, and Pontoporia, although its frontals do form the absolute apex just as they do in Inia, with a pedestal that can be directly pinched between an index finger and thumb, anterior of the apex of the supraoccipital shield. Notably, Isthminia lacks the strongly inflated bosses of the premaxillary sac fossae seen in nearly all other inioids (Figs. 5 and 6).

The mandible of Isthminia is most similar to Inia, in terms of an elongate mandibular symphysis, morphology in transverse section, and general size (Figs. 8 and 9). Both Isthminia and Inia lack the distinct ventrolateral groove in Pontoporia. Mental foramina with overhanging sulci are prominent in Isthminia, but smaller in Inia, although in both they extend posteriorly along the body of the ramus; also, the anterior termination of the mandibles in Isthminia is rounded in lateral view, whereas it is more angular in Inia. In lateral view, the coronoid process in Isthminia is less elevated, relative to the level of the trough in the mandibular symphysis than either Inia or Pontoporia. Both in Isthminia and Inia, the posterior termination of the dentition and the anterior termination of the acoustic window occur in close proximity, whereas in Pontoporia these landmarks are separated by a large gap along the mandibular ramus. Lastly, for the scapula, Isthminia shares the most similarities with Inia, although the scapula is not known in the majority of fossil inioids, and it remains unpublished in the otherwise abundantly represented Brachydelphis mazeasi (e.g., MUSM 887). We note the presence of both a complete scapula and a humerus in the type specimen of Incacetus broggii Colbert, 1944 (AMNH 32656) from marine strata in the Ica Desert (likely the Pisco Formation, although it may derive from older strata) in Peru. Both elements hint at inioid affinities for this taxon, from the Pisco Basin, which has previously been identified as a kentriodontid (Muizon, 1988b).

Taphonomy, body size, and ecomorphology

Isthminia was recovered from the type locality with the ventral surface of the skull exposed stratigraphic up, at the outcrop surface, directly overlying the mandibles, which were preserved slightly askew from the main axis of the skull, dorsal surface up (Fig. S1). Careful inspection of the surrounding quarry, prior to excavation, led to the recovery of 3 isolated teeth. The scapula was recovered within 1 m of the skull and jaws, mid-way through the excavation. Overall, the distribution of the skeletal elements at the type locality are similar to other fossil odontocetes in the same size, in similar depositional environments, and that have been recovered as associated skeletons (e.g., Tanaka & Fordyce, 2015). By comparison, there are generally far fewer postcranial elements preserved with the type specimen of Isthminia than might be expected, suggesting that most of the skeleton was likely eroded away from overlying rock.

The degree of disarticulation at the type locality corresponds to Articulation Stage 2 described by Pyenson et al. (2014) in their supplemental files, which matches the same articulation stage in Boessenecker, Perry & Schmitt (2014). In terms of bone modification, there is no evidence of bite marks from marine macroscavengers, and we did not observe any of the phosphatization, fragmentation and polish described by Boessenecker, Perry & Schmitt (2014) for marine vertebrates from the Mio-Pliocene age Purisima Formation of California. In sum, these observations point to the type specimen of Isthminia representing a single individual skeleton showing little transport, slight disarticulation, and burial in a low energy depositional environment.

Using both the Platanistoidea and Delphinoidea body size equations from Pyenson & Sponberg (2011), we calculated the total length of Isthminia between 284 and 287 cm, respectively, based on an estimate of the bizygomatic width of the skull by doubling the distance from the lateral surface of the zygomatic process to the midpoint of the mesethmoid. Assuming the type specimen represents a mature individual, this total length exceeds the largest values for Inia (LACM 19590 with TL = 221 cm) and Pontoporia (CAS 16529, with TL = 157 cm) from the adult specimens cited in Pyenson & Sponberg (2011)’s dataset, although we note that adult Inia and Pontoporia can attain lengths as large as 300 cm and 175 cm, respectively (Nowak, 1999). The reconstruction of Isthminia’s TL closely matches medium- to large-sized extant delphinoids, such as Grampus griseus (Cuvier, 1812), which has an average TL of 283 cm, based on 8 adult specimens in Pyenson & Sponberg (2011)’s dataset. Notably, Isthminia ranks among the largest of inioids, though it was slightly smaller than a similar estimate for Ischyrorhynchus (TL of 288–291 cm based on MACN 15135). Saurocetes spp., a Pan-Inia taxon, was likely much larger, but poorly known, based on incomplete material from the Ituzaingó Formation of Argentina for Saurocetes gigas (only known from a proximal fragment of a mandibular symphysis and isolated teeth), and mandibles and partial cranial specimens for S. argentinus from the late Miocene Ituzaingó, Urumaco, and Solimões formations of Argentina, Venezuela, and Brazil, respectively (see Gutstein, Cozzuol & Pyenson, 2014b).

We also examined two relevant morphological ecomorphological indices: mandibular bluntness index (MBI) and proportional orbit size. First, we followed methods outlined by Werth (2006) and calculated a MBI value of 0.548 for Isthminia, which is greater than values for either Inia or Pontoporia. By comparison, the MBI value for Isthminia most closely resembles those for other delphinids reported by Werth (2006). We also generated a simple metric to compare relative orbit size (ROS) among odontocetes, in an effort to better quantify the proportionally large orbits of Isthminia, especially with respect to Inia and Pontoporia. Using antorbital notch width to control for size (following Pyenson & Sponberg, 2011), we calculated a ROS value for Isthminia at 0.40 (Table 3). This value is larger than Inia, but smaller than Meherrinia, Pontoporia, and Brachydelphis spp. Comparisons among the dimensionless ROS indices do not immediately reveal any strong phylogenetic or ecologic structuring (Table 3), with Isthminia having a ROS in the same range as fossil and living marine odontocetes. It is entirely possible that ROS does not have the same importance in the sensory ecology of odontocetes as it does in other marine mammals that do not echolocate and therefore depend much more on visual prey detection (Schusterman et al., 2000; Debey & Pyenson, 2013).

The preponderance of occlusal wear facets on the apices of the lower and upper tooth crowns is not dissimilar from extant delphinioids, such as off-shore specimens of Tursiops, and fossil delphinidans such as Lophocetus pappus. Following Loch & Simões-Lopes (2013), we scored 100% dental wear in the type and only specimen of Isthminia, with every tooth showing at least superficial apical tooth wear of the dental crown; only 2 out of 41 complete teeth in the dentition of the type specimen showed simultaneous wear in apical and lateral tooth facets. Although the percentage of simultaneous apical and lateral wear ranks comparatively low for Loch & Simões-Lopes (2013)’s dataset of delphinids from the coast of Brazil, the dominance of superficial dental crown corresponding to Index 1 is generally in line with similar modes from pelagic delphinids in their dataset. We note, however, that Isthminia has different overall tooth morphology and lower tooth counts as compared with stem and crown delphinoids, and fewer teeth than Inia and Pontoporia.

Overall, Isthminia shares some ecomorphological similarities with pelagic odontocetes, especially with delphinioids of comparable body sizes and MBI. In comparative studies of congeneric or closely related odontocetes with disparate habitat preferences (e.g., riverine or estuarine versus oceanic settings), Monteiro-Filho, Monteiro & Reis (2002) and Galatius et al. (2011) demonstrated associated morphological complexes with each habitat. Pelagic odontocetes, for example, tend to have rostra with less ventral inclination, possess facial regions that were dorsally and posteriorly expansive and more concave in lateral aspects. Isthminia is consistent with this pelagic characterization in having a rostral plane less ventrally inclined than either Inia or Pontoporia, and having a broad exposure of the maxillae posterior of the level of the external nares. Future work should quantify these features more broadly across fossil delphinidans in a way that can be comparable with Galatius et al. (2011)’s findings.

Environmental and ecological implications

Planktotrophy is the dominant feeding mode of both the benthonic and nektonic fossil invertebrate assemblages preserved in the Piña Facies (Schwarzhans & Aguilera, 2013; O’Dea et al., 2007) demonstrating high planktonic productivity. In contrast, modern Caribbean shelf shallow waters are dominated by low planktonic but high benthic productivity driven by autotrophic reef and seagrass growth (O’Dea et al., 2007). The shift from planktotrophic to autotrophic benthic communities took place across the Caribbean when the Isthmus of Panama formed ∼3.5 Ma (Jackson & O’Dea, 2013). The presence of Isthminia and other predators including billfishes (Fierstine, 1978; J Velez-Juarbe, pers. comm., 2015), chondrichthyans (Carrillo-Briceño et al., 2015) and cetaceans such as kogiids (Velez-Juarbe et al., 2015), physeteroids (Vigil & Laurito, 2014), and delphinoids (J Velez-Juarbe, pers. comm., 2015), all with presumably high metabolic rates, corroborate further the presence of high planktonic productivity in the Piña Facies.

The source of high planktonic productivity is not yet resolved. Upwelled, nutrient-rich Pacific waters may have entered the Caribbean coast of Panama (O’Dea et al., 2012) through the remaining straits of the Central American Seaway (Jackson & O’Dea, 2013; Coates & Stallard, 2013; Leigh, O’Dea & Vermeij, 2014) in the late Miocene. High rates of cloning in cupuladriid bryozoans (O’Dea & Jackson, 2009), high variations in stable isotopes along skeletal profiles from gastropod shells (Robbins et al., 2012), and high variations in temperature-mediated zooid sizes (O’Dea et al., 2007) all suggest that strong seasonal upwelling was a dominate regime in this area. Alternatively, nutrients may have originated from more localized terrestrial runoff, as proposed for emergent platforms in present-day Colombia (Montes et al., 2015). However, reconciling the small watershed of the Isthmus of Panama with the geographic and stratigraphic extent of the Piña Facies (approximately 40–50 m thick) make it unlikely that high productivity levels observed throughout the facies could have been maintained solely from terrestrial input, even if higher rainfall and greater orogenic or volcanic activity in the late Miocene resulted in increased nutrient input from the proto-Isthmus. As such, it is unlikely that there were large rivers close to the area, further corroborating the hypothesis that Isthminia lived in a fully marine habitat.

The high abundance of benthic foraminifera assemblages with modern or ancient upper and middle bathyal depth ranges led Collins et al. (1996) to conclude that the Piña Facies of the Chagres Formation was deposited in deeper waters. Collins et al. (1996) suggested that the Piña Facies were preserved as the Central American Seaway deepened following the deposition of the underlying shallow-water Gatún Formation, and therefore represented the ephemeral formation of a fairly deep oceanic connection from the Pacific Ocean into the Caribbean Sea, prior to final closure of the Isthmus of Panama. This pattern of sediment deepening at the end of the Miocene, followed by shallowing and final closure of the Isthmus in the Late Pliocene, repeats itself across several basins along the Isthmus of Panama (Coates et al., 2003; Coates et al., 2004), pointing to pervasive regional eustatic sea level rise at the end of the Miocene (Miller et al., 2005) as a driver.

De Gracia et al. (2012) suggested that the extent of deepening at this time was extreme. They used the vast abundance of lanternfish (e.g., Diaphus Eigenmann & Eigenmann, 1890) recovered from the sediments (Schwarzhans & Aguilera, 2013) as evidence that the Piña Facies was deposited in up to 700 m of water depth (see File S2 for otolith abundance data from this unit, near the type locality). Although lanternfish do inhabit deeper waters during the day to avoid predation, they are well known to migrate into shallow waters at night to feed. Indeed, their otoliths are abundant in shallow water (<35 m) sediments in Bocas del Toro today. Thus, the presence of lanternfish, even in the great abundance observed in the Piña Facies is insufficient to assume deep-water deposition.

In a more recent study, Hendy et al. (in press) used molluscan, foraminferal, coral, and fish otolith assemblages, along with detailed sedimentological evidence, to conclude that the deepening event was considerably less pronounced. They suggested the deposition of the Piña Facies was around 125 m in depth, closely reflecting a previous estimate made by Collins et al. (1999) using corals and fish otoliths. Intense productivity or upwelling characteristic of the Piña Facies could have compressed thermoclines and compensation depths resulting in an apparent compression of the depth ranges of diagnostic taxa resulting in possibly anomalously deep estimates. The presence of a single specimen of Isthminia sheds little light on this palaeodepth discussion, except to note that modern day pelagic delphinoids concentrate around the neritic zone (Benoit-Bird & Au, 2003; Gowans, Würsig & Karczmarski, 2007; Benoit-Bird & McManus, 2012).

The evolutionary history of inioidea in the Americas

The fossil record of Inioidea reveals a far broader geographic distribution in the past than would be predicted from the extant ranges of Inia and Pontoporia. Fossil inioids outside of South America have predominantly been recovered from marine deposits representing nearshore depositional environments, although Isthminia’s recovery from rocks representing potentially a open ocean setting is consistent with ecomorphological traits that Isthminia shares with pelagic odontocetes alive today (Fig. 14). Although some freshwater Pan-Inia lineages from the late Miocene of Argentina may have been ∼4 m in total length, they are based on fragmentary remains (Cozzuol, 2010), and Isthminia is the largest marine inioid yet reported, in addition to being the only fossil inioid known from the Caribbean. Based on the available evidence, Isthminia occupied a high trophic level in a highly productive fully marine tropical Caribbean coastal ecosystem that predated the complete formation of the Panamanian Isthmus. Many of the bony fish species that are recorded in spectacular abundance from adjacent otolith assemblages in the Chagres Formation (File S2) may have formed a dominant portion of the prey resources for Isthminia, as they do for extant delphinids (see Kelley & Motani, 2015).

Figure 14 Reconstruction of Isthminia.

Life reconstruction of Isthminia panamensis, feeding on a flatfish, which would have been abundant in the neritic zone of the late Miocene equatorial seas of Panama. Art by Julia Molnar.

Hamilton et al. (2001) suggested that the marine ancestors of Inia, subsequent to their divergence from Pontoporia, invaded freshwater ecosystems of Amazonia during eustatic sea-level highs of the middle Miocene, and evolved freshwater habits prior to the subsequent drop in eustatic sea-level late in the Neogene. This proposed evolutionary scenario is entirely consistent with the late Miocene (Messinian) antiquity of Isthminia, which establishes a minimum boundary on its divergence with Inia (Fig. 15). Fossil remains attributable directly to Inia spp. have been reported from Pleistocene age freshwater deposits of the Rio Madeira Formation in Brazil (Cozzuol, 2010). An isolated Pan-Inia humerus from the late Miocene Ituzaingó Formation implies that this clade had already invaded turbid, obstructed shallow rivers and flooded forests typical of today’s Amazonian freshwater ecosystems by this time, although this humerus may belong to extinct taxon more closely related to Ischyrorhynchus (Gutstein et al., 2014a).

Figure 15 Stratigraphically calibrated phylogenetic tree of Inioidea.

Time calibrated phylogenetic tree of select Delphinida, pruned from our consensus cladogram in Fig. 13, including Isthminia, with Delphinoidea collapsed. Stratigraphic range data derives from published accounts for each taxon, including global ranges. Geologic time scale based on Cohen et al. (2013). Calibration for major nodes depths follow mean divergence date estimates by McGowen, Spaulding & Gatesy (2009: table 3) for the following clades: a, Delphinida (24.75 Ma); b, Inioidea + Lipotes (22.15 Ma); c, Delphinoidea (18.66 Ma); and Inioidea (in open white circle, 16.68 Ma). All minor node depths are graphical heuristics, and not intended to reflect actual divergence dates. Arc indicates stem-based clade, Pan-Inia. Ecological habitat preference is based on depositional environment or extant habitat. Abbreviations: Aquitan., Aquitanian; H., Holocene; Langh., Langhian; Mess., Messinian; P., Piacenzian; Ple., Pleistocene; Plioc., Pliocene; Serra., Serravallian; Zan., Zanclean.

The results of our phylogenetic analysis, however, cast some complexity on a simple scenario of marine-to-freshwater directionality given the phylogenetic placement of Ischyrorhynchus, from freshwater deposits of South America. Taken at face value, our analysis points to either two separate freshwater invasions in South America from marine ancestry at different times (one for Ischyrorhynchus, and another for Inia), or a single invasion with the origin at the unnamed clade of Ischyrorhynchus + Isthminia + Inia, with a marine re-invasion leading to Isthminia (Fig. 15). While the overwhelming marine ancestry for Inioidea is clear from the phylogenetic background of most odontocetes, there is no clear parsimonious argument for the directionality of marine-freshwater ecological transitions. Geisler et al. (2011) discussed such ecological complexity in considering Hamilton et al. (2001)’s scenario, pointing specifically to separate instances of overlapping geographic and ecological distributions between sympatric pairs of exclusively freshwater and estuarine to marine odontocete taxa: e.g., Sotalia fluviatilis (Gervais, 1853) with Sotalia guianensis (Van Beneden, 1864), both delphinids, in South America (Cunha et al., 2005; Caballero et al., 2007; Gutstein, Cozzuol & Pyenson, 2014b); and, prior to the former’s extinction, Lipotes vexillifer and Neophocaena phocoenoides (Cuvier, 1829), a phocoenid, in China. These extant examples, along with the recent fossil discoveries of putatively marine odontocetes in freshwater depositional environments (Bianucci et al., 2013; Boessenecker & Poust, 2015) suggest that freshwater invasions by marine odontocetes have happened frequently throughout the Neogene, in different continental margins, across major lineages, and, as our results suggest, perhaps within clades as well.

For South America, we conclude that marine odontocetes likely invaded freshwater ecosystems several times, with platanistids representing an initial invasion in the middle Miocene that ultimately disappeared, prior or subsequent to later a singular or repeated inioid invasions in the late Miocene. Future work, including new discoveries, will hopefully increase branch support for the phylogenetic arrangement of Pan-Inia (and basal inioids), and better refine this scenario for South American inioid evolution, and elsewhere. These evolutionary hypotheses may also be compared with diversification and selective extinction patterns for other vertebrate groups that invaded Amazonian freshwater ecosystems from marine ancestries (e.g., stingrays belonging to Potamotrygonidae Garman, 1877, see Lovejoy, Bermingham & Martin, 1998; croakers in the genus Plagioscion Gill, 1861, see Cooke, Chao & Beheregaray, 2012), in conjunction with the timing of orogenetic events during the late Neogene (Hoorn et al., 2010). Lastly, comparative phylogenetic analyses of the physiology and functional morphology of odontocetes, and other possible marine tetrapod analogs that have overlapping ecological occupancy will also provide a better basis for evaluating adaptational hypotheses that explain their evolution (Kelley & Pyenson, 2015).

Supplemental Information

Figure S1 Collecting the type specimen of Isthminia at the type locality on 18 June 2011.

(A) The specimen exposed in the outcrop, at low tide, with the scapula, oriented lateral side facing stratigraphic up, approximately 35 cm away from the skull, which was exposed ventral side up. Scale bar = 10 cm. Photo: J. Velez-Juarbe. (B) With the high tide returning, removal of the plaster jacketed sediment block, containing the skull, exposed the mandibles located directly underneath it. The mandible was oriented dorsal surface facing stratigraphic up. Photo: A. O’Dea.

Click here for additional data file.

File S1 File S1

Click here for additional data file.

File S2 Otolith data collected by Orangel Aguilera from the Chagres Formation.

Click here for additional data file.

Video S1 Supplemental Video 1

CT movie though the skull Isthminia panamensis, using a stack of DICOM images, processed in ImageJ (v. 1.48), oriented dorsal to image top, right lateral to the image right, and left lateral to image left. Original CT data are available at Smithsonian X 3D (http://3d.si.edu).

Click here for additional data file.

Foremostly, we thank Felix Rodriguez, Owen McMillan and Eldredge Bermingham for their support. We would also like to thank Celideth DeLeon for her circumspect administrative help. We are grateful to the Government of Panama’s Ministerio de Comercio e Industrias (MICI) and staff at STRI for assistance, including Santosh Jagadeeshan, Andrew Ugan and Carlos De Gracia, for logistical support with collecting and transporting the type specimen of Isthminia, which was collected and exported with permits from the MICI. We thank Steven Jabo and Peter Kroehler (USNM) for technical assistance with preparation of the type specimen. We also thank Tomas Hrbek, an anonymous reviewer, and Academic Editor Mark Young for helpful comments on the manuscript. For insightful discussions and access to unpublished data, we thank Stephen J. Godfrey, Olivier Lambert, Mizuki Murakami, R. Ewan Fordyce, James G. Mead, Charles W. Potter, David J. Bohaska, and Alexander J. Werth. We also thank James F. Parham for suggestions and comments on many drafts of the manuscript. We also thank David J. Bohaska, Charles W. Potter, John J. Ososky, and James G. Mead (all USNM), John R. Nance and Stephen J. Godfrey (CMM), Rodolfo S. Gismondi (MUSM), Christine Argot, Guillaume Billet, and Christian de Muizon (all MNHN), Patricia A. Holroyd (UCMP), Marcelo Ruguero (MLP), Alejandro Kramarz (MACN), and John Flynn, Ruth O’Leary, Nancy B. Simmons, and Neil Duncan (all AMNH), and staff at the Instituto de Desenvolvimento Sustentável Mamirauá in Tefé, Brazil, who all provided access to collections under their care. We are grateful for the support of NMNH Imaging, including the late Donald Hurlbert, James Di Loreto, Brittany Hance, and Kristen N. Quarles, for photography of the type specimen of Isthminia. We thank Tatjana Dzambazova (Autodesk), Antonije Velevski and Ralph Wiedemeier for website assistance with the 3D model of Isthminia, Ping Fu (3D Systems) and Günter Waibel, Adam Metallo, Vincent Rossi, and Jon Blundell of the Smithsonian’s Digitization Program Office for their support with 3D digitization. We are grateful to Esteban Pacheco for printing and donating a 3D model of Isthminia. We also thank Tina Tennessen for her timely editorial skills, and the input of Lauren O’Regan and Loretta Cooper (all at USNM). Orangel Aguilera kindly gave permission to use his fish otolith data from the Chagres Formation. Marcelo Viana and Orangel Aguilera kindly allowed us to use fish images in preliminary drafts of Isthminia’s reconstruction. We are indebted to the generosity and expertise of Laurel Collins, for help with the stratigraphy and age of the Chagres Formation, and we thank Austin Hendy for the base map in Fig. 2. Lastly, we thank Julia Molnar for her creative and careful execution on the life reconstructions of Isthminia.

Anatomical Abbreviations

adif anterior dorsal infraorbital foramen

alis alisphenoid

ap angular process of mandible

C canine tooth

cp coronoid process of mandible

cuneif cuneiform

dsss dorsal sagittal sinus sulcus

fplpts fossa for the postorbital lobe of the pterygoid sinus

fr frontal

gf glenoid fossa of squamosal

I incisor tooth or teeth

ju jugal

la lacrimal

Ma mega-annum, period of 1 million years

max maxilla

mc maxillary crest

me mesethmoid

mef mental foramen or foramina

mf mandibular foramen

ms mandibular symphysis

na nasal

nar bony narial opening or naris

nuc nuchal crest

pa parietal

PC postcanine tooth or teeth

pdif posterior dorsal infraorbital foramen

pls posterolateral sulcus of the premaxilla

pmax premaxilla

pmaxf premaxillary foramen

pms posteromedial sulcus of the premaxilla

pmsf premaxillary sac fossa

popf postorbital process of the frontal

propf preorbital process of the frontal

scap scaphoid

socc supraoccipital

sopf supraorbital process of frontal

smf suprameatal fossa

sq squamosal

tc temporal crest of the frontal

trap trapezoid

uncif unciform

vom vomer

zpsq zygomatic process of squamosal

Institutional Abbreviations

AMNH Divisions of Paleontology and Vertebrate Zoology, American Museum of Natural History, New York, New York, USA.

CAS Department of Ornithology and Mammalogy, California Academy of Sciences, San Francisco, California, USA.

CMM Calvert Marine Museum, Solomons, Maryland, USA.

IRSNB Institut royal des Sciences naturelles de Belgique, Brussels, Belgium.

LACM Departments of Mammalogy and Vertebrate Paleontology, Natural History Museum of Los Angeles County, Los Angeles, California, USA.

MACN Museo Argentino de Ciencias Naturales “Bernardino Rivadavia,” Buenos Aires, Argentina.

MLP Museo de La Plata, La Plata, Argentina.

MNHN Muséum national d’Histoire naturelle, Paris, France.

MUSM Museo de Historia Natural, Universidad Nacional Mayor San Marcos, Lima, Peru.

UCMP University of California Museum of Paleontology, Berkeley, California, USA.

UF Florida Museum of Natural History, Gainesville, Florida, USA.

USNM Departments of Paleobiology and Vertebrate Zoology (Division of Mammals), National Museum of Natural History, Smithsonian Institution, Washington, D.C., USA.

Additional Information and Declarations

Competing Interests

Author Contributions

Field Study Permissions

Data Availability

New Species Registration

Nicholas D. Pyenson is an Academic Editor for PeerJ.

Nicholas D. Pyenson, Jorge Vélez-Juarbe, Carolina S. Gutstein and Aaron O’Dea conceived and designed the experiments, performed the experiments, analyzed the data, contributed reagents/materials/analysis tools, wrote the paper, prepared figures and/or tables, reviewed drafts of the paper.

Holly Little performed the experiments, analyzed the data, wrote the paper, prepared figures and/or tables, reviewed drafts of the paper.

Dioselina Vigil conceived and designed the experiments, performed the experiments, reviewed drafts of the paper.

The following information was supplied relating to field study approvals (i.e., approving body and any reference numbers):

Fossil material was collected and exported under the Government of the Republic of Panama’s Ministerio de Comercio e Industrias (MICI number DNRM-MC-074-11).

The following information was supplied regarding the deposition of related data:

Full resolution 3D models and original CT data are available online at Smithsonian X 3D (http://3d.si.edu) and archived, along with supplemental data, in Zenodo (https://zenodo.org/record/27214) at the following DOI: 10.5281/zenodo.27214.

The following information was supplied regarding the registration of a newly described species:

Isthminia

urn:lsid:zoobank.org:act:83F6A9B4-289D-45DE-A3D1- C361DAAAF973

Isthminia panamensis

urn:lsid:zoobank.org:act:A5C706B6-E0B6-43E5-A65C-E6FE0B2BDF1A.

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
