# Peer review of "Isthminia panamensis, a new fossil inioid (Mammalia, Cetacea) from the Chagres Formation of Panama and the evolution of ‘river dolphins’ in the Americas"

_PeerJ, doi:10.7717/peerj.1227_

## Round 0.1 · original submission · Major Revisions

As the two reviewers have made differing recommendations, I have made the decision of 'major revision'.

I have some additional comments that the authors should address prior to resubmission (in addition to those made by the reviewers):

1. Authority and date should be provided for each species-level taxon at first mention. Please ensure that the nominal authority is also included in the reference list.
2. Please replace 'trees' with 'cladograms' where appropriate.
3. It is the ensemble consistency index and ensemble retention index, not consistency and retention indices.
4. The clade 'Pan-Iniidae'. This clade name does not follow the Code of the ICZN. If creation of a new clade is necessary (see reviewer one on how do pan-iniids and iniids differs), could the authors establish the clade name under the ICZN Articles.

Reviewer 1 ·

Basic reporting

No comments

Experimental design

No comments

Validity of the findings

No comments

Additional comments

This manuscript is an important and valuable contribution into the evolutionary history of river dolphins. A new Inia-like genus and species from a marine habitat in the Caribbean region is described which would drive the further research in the taxonomy and ecology of inioids.

However, a few issues would be addressed before the publication. First, the phylogenetic affinities of the new genus are now uncertain: it is unclear if the new genus is closer to Inia or Pontoporia. The clear differential diagnosis of the new genus is necessary. The phylogeny is now based on the analysis by Aguirre-Fernández and Fordyce (2014) which was originally conducted for stem odontocetes but not for resolving the relationships within the crown groups (for example, it grouped Platanista with Lipotes and separately from Zacharias; Atocetus with Albireo; Orcinus with Globicephala; Leucopleurus with Delphinus, etc.); thus, the phylogenetic analysis should be run on another matrix. Second, the reported 3D model of the specimen would be used in the description or discussion of relevant aspects (hearing, brain anatomy or geometric morphometry). Third, the discussion of ecology would be based on the anatomy, microanatomy or chemistry of the specimen.

Other specific comments are provided below.

Lines 62–66: “there is poor phylogenetic resolution for the placement of Lipotes and Platanista among basal branching lineages of Odontoceti (Messenger & McGuire, 1998; Hamilton et al., 2001; Nikaido et al., 2001; Geisler & Sanders, 2003; Arnason et al., 2004; May-Collado & Agnarsson, 2006; Steeman et al., 2009; Geisler et al. 2011).”

Lipotes and Platanista were pooled together in none of the papers cited here, except the morphological analysis by Geisler & Sanders, 2003. On the contrary, all known molecular studies (including Yan et al., 2005; Price et al., 2005; McGowen et al., 2009), concurred in placing Platanista outside of Delphinida, and Lipotes within Delphinida. This was also confirmed by combined molecular and morphological analyses (Geisler et al., 2011; Gatesy et al., 2012). Lipotes was grouped with Platanista only in a minority of purely morphological analyses.

Lines 135–140: The details why the CT scans were used instead of the surface scans are unnecessary and can be omitted.

Lines 152–161: It is unclear why the authors did not use the referred matrix by Geisler et al. (2012) which is focused on Inioidea and is better sampling and better resolving than the matrix by Aguirre-Fernández & Fordyce (2014): the latter one was constructed for a stem Odontoceti taxon, and it can be hardly applied for crown groups.

Lines 168–169: Noteworthy, both Geisler et al. (2012) and Aguirre-Fernández & Fordyce (2014) used a variety of cladistic algorithms and the TNT software (Goloboff et al., 2008). The different results can be partly explained by the method of analysis.

Line 181: It is unclear what is the difference between Pan-Iniidae and Iniidae. Please provide the diagnosis for Pan-Iniidae and the diagnosis for Iniidae differing Inia from the other Pan-Iniidae.

Lines 206–208: Formation and age: Please provide the brief characteristics of the locality, including the features (organisms) used for the dating.

Lines 212–213: The diagnostic traits of Odontoceti are unnecessary here.

Lines 213–215: The mandibular and lacrimal characters mentioned here are plesiomorphic.

Line 222: The left premaxilla is not preserved, making the diagnostic character unidentifiable.

Lines 253-261: The differential diagnosis, and particularly the differential diagnosis from Inia, is necessary.

Lines 288-289, ff.: shallow _longitudinal_ canals; please provide all the relevant labels on the illustrations

Line 301, ff.: please transfer all the comparative data and comparisons into the Discussion.

Lines 321–328: This noteworthy anatomy should be mentioned in the diagnosis. An additional close-look illustration can be useful.

Line 375: Antorbital fossa (see Mead and Fordyce, 2009)? Please provide all the relevant labels on the illustrations.

Line 403: “the postorbital process is more elongated with a triangular cross section”: it is unclear what plane of cross-section is meant.

Lines 404–411: The comparison with the orbit of Inia obviously relates to the freshwater habitat adaptations and thus is highly relevant as a part of Discussion.

Line 459: external occipital crest is not the sagittal crest

Lines 466–481: The anatomy of squamosal can be an important feature in which the new genus differs from Inia, and it worth for the more detailed description and discussion. The comparative illustrations of REDACTED and Inia will be helpful.

Line 664: “The recovery of Platanista in a close relationship with other lipotids has
been a frequent result of exclusively morphological analyses”: Please provide the full list of relevant references.

Line 672: In dorsal view, the described skull is more similar to Brachydelphis (rather than to Inia) in a wide and posteriorly shifted facial region.

Line 680–682: “In some ways, the rostrum of REDACTED is reminiscent of Kampholophos serrulus, from the late Miocene of California (Rensberger, 1969), which is likely a pan-iniid-mimic delphinoid,”: please provide all the relevant references and data, especially regarding the phylogeny of Kampholophos.

Lines 686–688. This trait, the anterior sinus invading the maxilla, is shared by Brachydelphis (Lambert and Muizon, 2013). In general, the comparison with Brachydelphis should be considered.

Lines 768–825: As admitted here, “The presence of a single specimen of REDACTED sheds little light on this palaeodepth discussion”. The single finding of a dead pelagic dolphin in a pelagic habitat gives nothing in terms of its ecology or distribution. It is highly recommended to omit this section as a whole.

Line 833: Please provide the discussion on the ecomorphological traits shared by pelagic odontocetes, with the relevant references. For example, the large number of teeth is the evidence for the fish diet; the prominent temporal fossa and large teeth indicate taking the large prey; the upside deflected rostrum is the evidence for pelagic (vs benthic) lifestyle, etc. See also Monteiro-Filho et al., 2002; Galatius et al., 2010, etc.

Lines 837-840: This statement is not supported by the diet analysis (for example, stomach or isotope contents or dental wear). The presence of otoliths at the place of the carcass burial does not mean this place was its feeding habitat.

·

Basic reporting

No Comments

Experimental design

No Comments

Validity of the findings

No Comments

Additional comments

Ln 730-744 – when discussing and comparing the size of REDACTED to other Inioid and Delphinoid taxa, the authors only use skeletal material in collections used in previous analyses. I realized that this makes for easy comparison, and the only comparison that is possible with extinct taxa, but the reported values subestimate the actual sizes of all the species of Inia and Pontoporia reported in literature. This should be acknowledge, and the reliance on the collection material should justified, if the authors choose to continue so.


Ln 865-866 – correctly said, co-occurances are between species (only species can interact with each other)

Figure 12 – it remains unclear to me how the time callibrated phylogeny was estimated. Was the phylogeny in Figure 10 pruned, and then node depth adjusted to the dates given in the literature cited in the Figure 12 legend?

Data matrix – the data matrix only lists character states of the 331 characters and the 52 taxa used in this study. However, it does not list what these characters are. Please provide a list of the characters used as well, and explanation of the character states.

Finally, and most importantly, please include an explicit diagnosis section for the new taxon. What are the diagnostic characters (autapomorphies) that unambiguously diagnose and differentiate this taxon from its relatives? If there are no diagnostic characters or character states, is there a combination of characters that is diagnostic? If so, what is that.

Additionally, what are the synapomorphies that place it in the sister group position to Inia spp.?

Are any of these apomorphies reversed anywhere else on the phylogeny?

---

## Round 0.2 · Minor Revisions

Dear authors,

While both reviewers have made a recommendation of 'accept', I have made the decision of 'minor revision'. This is due to the comments of reviewer two made in regards to the diagnosis. His comments should be quick and easy to make, and once done so I can foresee no reason not to accept the manuscript.

Reviewer two also mentions typos in the manuscript. Can the authors make one final read through of the manuscript as well.

Reviewer 1 ·

Basic reporting

No Comments

Experimental design

No Comments

Validity of the findings

No Comments

Additional comments

The revised version of the manuscript is significantly improved, especially in terms of stratigraphy and ecology. The arguments supporting the authors' views on taxonomic position and phylogeny of the new genus still seem to be challengeable. However, this is a subjective view which cannot impede the publication.

·

Basic reporting

No comments, everything appears to be OK.

Experimental design

No comments

Validity of the findings

No comments

Additional comments

I believe the authors successfully addressed all my comments and those of the other reviewer. I am happy with the form that the MS is in now, but see rest of the comments.

With respect to the diagnosis of the new genus and species, both the other reviewer and I asked for an explicit diagnostic section, which suggests that for both us the diagnosis was unclear, although, as the authors point out, the MS contains a three page diagnosis. I carefully read though this section again, and I think it was unclear for several reasons including that the diagnosis is of both the species and the genus, and the diagnosis is also a statement of the phylogenetic relationship of the new species, establishing that it belongs to the Pan-Inia clade, and that it is sister taxon to Inia.

Thus while I recommend accept, I suggest that the diagnosis is restructure in the final version of this MS in a way I believe will make it easier to follow. I suggest this order:
1) The first paragraph L279-285 is kept (establishing that the new species belongs in Inioidea)
2) Second paragraph L323-331 establishes that this species is distinct and diagnosable via autapomorphies from all other Iniodea species (currently the last paragraph of the diagnosis)
3) Third paragraph L287-306 establishes through combination of characters that it is not a member of one of the existing genera (currently the second paragraph of the diagnosis)
4) Fourth paragraph L308-321 establishes that the new species is within the Pan-Inia clade, and is sister to Inia (currently the third paragraph of the diagnosis - this section in a strict sense is not a diagnosis, and this perhaps should be indicated via a heading or sub-heading such as "Phylogenetic relationships to other Inioidea taxa")

I also realize that given that most of the genera are monotypic, sections 2 and 3 in practice diagnose the same thing, that the new species is distinct from all others, but given that section 2 lists autapomorphies while section 3 lists combinations of characters, from a diagnostic point of view, autapomorphies provide better diagnoses, and in my opinion should be listed first.

Last there are few typos (Cooke et al. 2011 in the text and Cooke et al. 2012 in the bibliography; L1801 "heursitics"; and few others which I am sure will be fixed during copy editing)

---

## Round 0.3 · accepted · Accept

Dear authors,

Thank you for your timely response to the revision. I am recommending 'accept' for your manuscript.

However, in proof-stage can you double-check trees v. cladograms? I noticed some instances in which cladogram would be more appropriate.